KCL-MTH-21-01

# Four-Dimensional Chern-Simons and Gauged Sigma Models

Jake Stedman[*][1]

[1]Department of Mathematics, King's College London,
Strand, London, WC2R 2LS, UK

March 8, 2023

## Abstract

In this paper we introduce a new method for generating gauged sigma models from four-dimensional Chern-Simons theory and give a unified action for a class of these models. We begin with a review of recent work by several authors on the classical generation of integrable sigma models from four dimensional Chern-Simons theory. This approach involves introducing classes of two-dimensional defects into the bulk on which the gauge field must satisfy certain boundary conditions. One finds integrable sigma models from four-dimensional Chern-Simons theory by substituting the solutions to the equations of motion back into the action. The integrability of these sigma models is guaranteed because the gauge field is gauge equivalent to the Lax connection of the sigma model. By considering a theory with two four-dimensional Chern-Simons fields coupled together on two-dimensional surfaces in the bulk we are able to introduce new classes of 'gauged' defects. By solving the bulk equations of motion we find a unified action for a set of genus zero integrable gauged sigma models. The integrability of these models is guaranteed as the gauge fields remain gauge equivalent to Lax connections. Finally, we consider two examples in which we derive the gauged Wess-Zumino-Witten and nilpotent gauged Wess-Zumino-Witten models. This latter model is of note as one can find the conformal Toda models from it.

[*]jake.williams@kcl.ac.uk

# 1 Introduction

Over the last two decades, several groups have turned their focus to the question of whether one can use gauge theories to identify properties of conformal field theories (CFTs), vertex operator algebras, and integrable models. We know of three such examples: the first, by Fuchs et al in [1–5], uses topological field theories to analyse conformal field theories. The second, by Beem et al, has shown a deep relationship between $\mathcal{N} = 2$ superconformal field theories in four dimensions and vertex operator algebras [6, 7]. The final example began with the work of Costello in [8, 9] and has since been expanded upon by Costello, Witten, and Yamazaki in [10–12]. In this series of papers the authors introduced a new gauge theory, called four-dimensional Chern-Simons theory (4d CS), and used it to explain several properties of two dimensional integrable models. In [10, 11] the authors were able to find the $R$-matrix and Quantum group structure of lattice and particle scattering models from Wilson lines on 4d CS theory. A fourth paper in this series [13], has also shown 't-Hooft operators are related to $Q$-operators.

We are interested in the third paper [12] in which the authors proved that classical four-dimensional Chern-Simons theory with suitably chosen two-dimensional defects reduces to a two-dimensional integrable sigma model. One finds a sigma model by solving the equations of motion in terms of a group element $\hat{g}$, which is the model's field. The dynamical degrees of freedom of the sigma model are the values of this group element on certain defects. Integrable sigma models are of particular interest given they exhibit many of the phenomena present in non-abelian gauge theories, such as confinement, instantons or anomalies [14–17] while their integrability ensures they are exactly solvable [18–21].

This result was extended by Bittleston and Skinner in [22][1] where it was shown that higher dimensional Chern-Simons models can be used to generate higher dimensional integrable sigma models.

These constructions are analogous to the construction of Wess-Zumino-Witten (WZW) model as the boundary theory of three-dimensional Chern-Simons given in [23]. However, what makes these constructions different is that these models sit on defects in the bulk rather than on the boundary.

Alongside these developments, Vicedo, in [24], observed that the gauge field $A$ of four-dimensional Chern-Simons theory can be made gauge equivalent to the Lax connection $\mathcal{L}$ of the two-dimensional integrable sigma model ( with or without a spectral parameter). This result was expanded upon in [25] by Delduc, Lacroix, Magro and Vicedo (DLMV) where they construct a general action for genus zero integrable sigma models called the unified sigma model action. This result is remarkable for two reasons: the first is that the Lax connection of an integrable sigma model can be found geometrically by solving the equations of motion of four-dimensional Chern-Simons theory; and the second is that it gives a general action from which the actions in this class of sigma models can be found if their Lax connections are known. We will refer to this construction as the DLMV construction throughout this paper.

In all of this work, the inability to generate gauged sigma models whose target spaces are cosets (manifolds of the form $G/H$ where $G$ and $H \subseteq G$ are groups), other than symmetric space models [12], has been mentioned several times. Gauged sigma models are of particular interest given they include the GKO constructions [26–28] from which one can possibly find all rational conformal field theories (RCFTs).

The main result of this paper is to prove that one can generate coset sigma models by coupling together two four-dimensional Chern-Simons theories on new classes of two dimensional defects which are collectively called 'gauged defects'. We call this theory doubled four-dimensional Chern-Simons theory. This theory consists of two 4d CS theories, one for a group $G$ with gauge field $A$, and a second for a group $H \subset G$ with gauge field $B$; the two models are coupled only on the gauged defects. This result is analogous to the work of Moore and Seiberg in [29] where it was shown the GKO constructions are the boundary theory of a doubled three-dimensional Chern-Simons model - see also [30].

As before, $A$ and $B$ are gauge equivalent to two-dimensional Lax connections $\mathcal{L}_A$ and $\mathcal{L}_B$ on the defects (with or without spectral parameters), and the equations of the motion of the model are the Lax equations

---

[1]In this paper the process of solving the equations of motion is referred to as solving along the fibre.

for $\mathcal{L}_A$ and $\mathcal{L}_B$, together with "boundary conditions" on the defects. By following arguments similar to those made by Delduc et al in [25] we find a unified gauged sigma model from which a large class of integrable gauged sigma models can be found.

It could have been expected that a gauged 2d WZW model can be found from doubled four-dimensional Chern-Simons theory. Firstly, the gauged WZW model can be found from the difference of two WZW models (see appendix D) each of which can be found from four-dimensional Chern-Simons theory. The second reason is that four-dimensional Chern-Simons theory is T-dual to three-dimensional Chern-Simons, as was shown by Yamazaki in [31]. Hence, since the GKO constructions are the boundary theory of a doubled three-dimensional Chern-Simons, it is natural to expect that they can be found in four-dimensional Chern-Simons theory.

The structure of this paper is as follows: in section 2 we define four-dimensional Chern-Simons theory, deriving its equations of motion and boundary conditions amongst other properties. In section 3 we review the construction of integrable sigma models by Delduc et al in four-dimensional Chern-Simons theory. In this construction the authors solve four-dimensional Chern-Simons theory's equations of motion and substitute them back into the action; where they differ from Costello et al is in the choice of gauge in which they do these calculations. In section 4 we define the doubled Chern-Simons theory, deriving the gauged defects and describing its gauge invariance.

In section 5 we use the DLMV approach to derive the unified gauged sigma model and construct the normal and nilpotent gauged WZW models. These examples are notable for two reasons: the first is that the normal gauged WZW model gives an action for the GKO constructions as described in [32–36]; the second reason is that the Toda fields theories can be found from both of these action. In the former case this is as a quantum equivalence with the $G_k \times G_1/G_{k+1}$ GKO model, as shown in [37], while in the latter case this is proven via a Hamiltonian reduction as shown in [38]. It was also shown in [38] that one can find the $W$-algebras from the nilpotent gauged WZW model.

In section 6 we summarise our results and comment on a few potential directions of this research.

## 2 The Four-Dimensional Chern-Simons Theory

In this section we define the four-dimensional Chern-Simons theory and derive the equations of motion, give the various defect boundary conditions that we use and describe the gauge-invariance of the action.

### 2.1 The Action

4d CS theory is a theory of a gauge field $A = A_\mu dx^\mu$. It is defined on a four-dimensional manifold of the form $\Sigma \times C$ where $\Sigma$ and $C$ are each two-dimensional surfaces. Further, $C$ is also a complex manifold with coordinate $z$ equipped with a meromorphic 1-form $\omega = \varphi(z)dz$ with $\varphi(z)$ a rational function.

The Chern-Simons three form is

$$CS(A) = \left\langle A, dA + \frac{2}{3} A \wedge A \right\rangle, \tag{2.1}$$

and the action of the 4d CS theory is the integral

$$S_{4\mathrm{dCS}}(A) = \frac{1}{2\pi\hbar} \int_{\Sigma \times C} \omega \wedge \left\langle A, dA + \frac{2}{3} A \wedge A \right\rangle, \tag{2.2}$$

The gauge field $A$ is a connection on a principal bundle over the four-dimensional manifold $M = \Sigma \times C$, with complex Lie group $G_\mathbb{C}$. We take $\langle \cdot, \cdot \rangle$ to be a non-degenerate symmetric bilinear form proprtional to

the killing form of the complex Lie aglebra $\mathbf{g}_{\mathbb{C}}$. The gauge field $A_\mu = A_\mu^a T^a$ is in the adjoint representation[2] of $\mathbf{g}_{\mathbb{C}}$, and basis elements $T^a$ are normalised such that $\langle T^a, T^b \rangle = \delta^{ab}$. Note, when both entries in $\langle \cdot, \cdot \rangle$ are differential forms there is an implicit wedge product between the two entries. We will discuss two classes of four-dimensional Chern-Simons action in the following and refer to (2.2) as the standard action, or theory, for short.

By analogy with three-dimensional Chern-Simons, we call $\hbar$ the 'level'; although irrelevant to classical four-dimensional Chern-Simons, $\hbar$ will be relevant in section 4 when we introduce a second four-dimensional Chern-Simons field $B$.

While $\Sigma$ can be more general, we will usually take it to be $\mathbb{R}^2$. Generically, the theory does not depend on a metric on $\Sigma$ but we find it helpful to write the coordinates on $\Sigma$ as $x^\pm$ as the result of the construction is a two-dimensional Lorentzian theory in which $x^\pm$ are "light-cone coordinates".

The integrable models one can generate using four-dimensional Chern-Simons depend not only upon the choice of $G$, but also the choice of the complex surface $C$ and 1-form $\omega$. If $C$ is a Riemann surface of genus $g$, the Riemann-Roch theorem states that the number $n_z$ of zeros and $n_p$ of poles (counted with multiplicity) of $\omega$ satisfy the equation:

$$n_z - n_p = 2g - 2 \,. \tag{2.3}$$

In this paper we will only discuss genus zero integrable field theories and follow [12] by fixing $C = \mathbb{C}P^1$ with the coordinates $z$ and $\bar{z}$. We will also only consider forms $\omega$ with at most double poles. The presence of poles and zeros of $\omega$ requires the consideration of the behaviour of the field $A$ near these locations, which we call "boundary conditions". A point in $\mathbb{C}P^1$ is a two-dimensional surface in $\Sigma \times \mathbb{C}P^1$, therefore the poles and zeros of $\omega$ and their boundary conditions define two-dimensional defects in the theory. We call the defects associated to zeros type A defects and those associated to poles, type B defects. We discuss these in sections 3.4 and 2.3 respectively.

Let $P$ denote the set of all poles and $P^{\text{fin}}$ denote the set of poles of $\omega$ not including infinity, then $\omega$ on $\mathbb{C}P^1$ is of the form:

$$\omega = \eta_\infty dz + \sum_{q \in P^{\text{fin}}} \sum_{l=0}^{n_q - 1} \frac{\eta_q^l}{(z-q)^{l+1}} dz \,, \qquad \text{where} \qquad \eta_q^l = \text{res}_q \left( (z-q)^l \varphi(z) \right) \,, \tag{2.4}$$

and $\eta_\infty^1 = \text{res}_\infty(\omega/z)$ while $n_q$ denotes the order of the pole $q$. We have restricted ourselves to at most doubles poles, including at $\infty$, which is why $\omega$ contains no polynomials of $z$.

Let $Z$ denote the set of all zeros and let $m_\zeta$ be the multiplicity of the zero at $\zeta \in Z$.

We clearly have

$$n_p = \sum_{q \in P} n_q \,, \quad n_z = \sum_{\zeta \in Z} m_\zeta \,. \tag{2.5}$$

Furthermore, since $g = 0$ we have $n_p = n_z + 2$ and it follows that the number of poles (counted with multiplicity) in $P$ is always greater than $n_z$, which could be zero.

Finally, we will only consider $\omega$ with a first or second order pole at infinity. Since $n_p = n_z + 2$, $\omega$ must have at least one pole. If there is no pole at infinity, we pick the location of a pole: $z = q$, say. Noting then that the isometry group of the Riemann sphere consists of Möbius transformations, one can send the pole $q$ to infinity by the transformation $z \to 1/(z-q)$ (since inversions and translations are Möbius transformations). From here on, we assume $\omega$ has a pole at infinity and $P = P^{\text{fin}} \cup \{\infty\}$.

Before deriving the equations of motion we emphasise two important facts. The first is that although (2.2) is constructed from wedge products, and contains no metric, one might reasonably expect (2.2) to be invariant under all diffeomorphisms, or in the vernacular 'topological', this is not the case. This is because

---

[2]One should note that this is only possible when the adjoint representation is non-trivial. If the adjoint representation is degenerate, such as for $U(1)$, then one must use an alternative representation.

$\varphi(z)$ does not transform as a vector meaning $\omega$ is not topological, unlike $A$, thus the action is not topological but 'semi-topological' as it is invariant under all diffeomorphisms of $\Sigma$.

The second fact is that (2.2) has an unusual gauge invariance. The $z$ components of the gauge field and the exterior derivative $d$, $A_z dz$ and $dz\partial_z$, both fall out of (2.2) because $\omega = \varphi(z)dz$ and $dz \wedge dz = 0$. This means the action has the additional gauge invariance:

$$A_z \longrightarrow A_z + \chi_z \,, \tag{2.6}$$

where $\chi_z$ can be any $\mathbf{g}_{\mathbb{C}}$ valued function. As a result of this gauge invariance, all field configurations of $A_z$ are gauge equivalent which allows us to set $A_z = 0$ and in the following the gauge field $A$ is:

$$A = A_\Sigma + \bar{A}\,, \tag{2.7}$$

where $\bar{A} = A_{\bar{z}}d\bar{z}$ and $A_\Sigma = A_+ dx^+ + A_- dx^-$ is the restriction of $A$ to $\Sigma$.

We will also drop the $dz$ terms from the exterior derivative $d$. In general, one has

$$d = d_\Sigma + \bar{\partial} + \partial \tag{2.8}$$

where $d_\Sigma$ is the exterior derivative on $\Sigma$ and $\bar{\partial}$ and $\partial$ are the Dolbeault operators on $C$. However, we can drop the $\partial$ term since the exterior derivative $d$ appears either in the action containing $\omega$ (and $dz \wedge dz = 0$) or in gauge transformations of $A$ (and we will always set $A_z$ to zero). By an abuse of notation, we will also write this modified exterior derivative simply as $d$,

$$d = d_\Sigma + \bar{\partial}\,. \tag{2.9}$$

We note that $\partial$ and $\bar{\partial}$ act on functions as $dz\partial_z$ and $d\bar{z}\partial_{\bar{z}}$ respectively.

Finally, we give several identities which will be used extensively below. The derivations of these equations can be found in appendix A. Let $\Lambda = f(z)d\bar{z} \wedge dx^+ \wedge dx^-$ where the function $f(z)$ is assumed to be meromorphic with poles only at the zeros of $\omega$. An example of such a 3-form is $\mathrm{CS}(A)$ in the gauge $A_z = 0$. The top form $\omega \wedge \Lambda$ can be put into the form:

$$\omega \wedge \Lambda = \eta_\infty^1 z\partial\Lambda + \sum_{q \in P^{\mathrm{fin}}} \sum_{l=0}^{n_q-1} \frac{\eta_q^l}{l!} \frac{dz}{z-q} \wedge \partial_z^l \Lambda + \partial\psi\,, \tag{2.10}$$

where:

$$\psi = -\eta_\infty^1 z\Lambda + \sum_{q \in P^{\mathrm{sing}}} \sum_{l=1}^{n_q-1} \sum_{r=0}^{l-1} \frac{(-1)^{l-r}}{r!(l-r)!} \partial_z^{l-r-1} \left( \frac{\eta_q^l}{z-q} \partial_z^r \Lambda \right)\,. \tag{2.11}$$

while $P^{\mathrm{sing}} \subseteq P^{\mathrm{fin}}$ is the set of poles whose multiplicity is greater than one; $\partial\psi$ is singular at these poles. There are two ways in which the singular part $\partial\psi$ can be removed from the action ensuring it is regular. The first method is to subtract $\partial\psi$ from the action as was used in [39]. The second, implicitly used in [12] as well as in the following, is to restrict the bundles we consider to those satisfying boundary conditions such that $\psi$ is regular. This ensures integrals of $\partial\psi$ vanish since $\mathbb{C}P^1$ is compact.

It is simple to show the second term on the right-hand side of (2.10) is regular by changing coordinates to polar coordinates centred on $q$. Since $\Lambda$ contains $d\bar{z}$ it follows that $dz \wedge \partial_z^l \Lambda$ is proportional to $dz \wedge d\bar{z} = 2ir dr \wedge d\theta$ thus canceling the pole from $z - q = re^{i\theta}$. The same argument applies to the first term if one works in the coordinates $z = r^{-1}e^{-i\theta}$.

Our second identity lets us localise our action to the defects at the poles of $\omega$, it is:

$$I = \int_{\Sigma \times \mathbb{C}P^1} \omega \wedge \bar{\partial}\xi = 2\pi i\eta_\infty^1 \int_{\Sigma_\infty} \partial_z\xi + 2\pi i \sum_{q \in P^{\mathrm{fin}}} \sum_{l=0}^{n_q-1} \frac{\eta_q^l}{l!} \int_{\Sigma_q} \partial_z^l\xi\,. \tag{2.12}$$

To derive this equation we have sent $\bar\partial(\omega \wedge \xi)$ to zero, which we can do because $\omega \wedge \xi$ is meromorphic in $z$ and $\mathbb{C}P^1$ is compact, and have sent $\int_{\Sigma \times \mathbb{C}P^1} \partial\tilde\psi$ to zero where:

$$\tilde\psi = 2\pi i \eta_\infty^1 \delta^2(1/z) d\bar z \wedge \xi - 2\pi i \sum_{q \in P^{\mathrm{sing}}} \sum_{l=1}^{n_q-1} \sum_{r=0}^{l-1} \frac{(-1)^{l-r}}{r!(l-r)!} \eta_q^l \partial_z^{l-r-1} \left( \delta^2(z-q) d\bar z \wedge \partial_z^r \xi \right) . \tag{2.13}$$

Note, we can write and evaluate (2.12) as a sum over residues. Let $V_q \in \mathbb{C}P^1$ denote an open region which contains only the pole $q$, thus we have:

$$\begin{aligned}
\int_{\Sigma_q} \mathrm{res}_q \left( \omega \wedge \xi \right) &= \int_{\Sigma \times V_q} d\bar z \wedge \delta^2(z-q) \partial_z^{n_q-1} \left( \frac{(z-q)^{n_q}}{(n_q-1)!} \omega \wedge \xi \right) \\
&= \sum_{l=0}^{n_q-1} \int_{\Sigma \times V_q} d\bar z \wedge \delta^2(z-q) \binom{n_p-1}{l} \partial_z^{n_q-l-1} \left( \frac{(z-q)^{n_p-l}}{(n_p-1)!} (z-q)^l \omega \right) \wedge \partial_z^l \xi \\
&= \sum_{l=0}^{n_q-1} \frac{\eta_q^l}{l!} \int_{\Sigma_q} \partial_z^l \xi , 
\end{aligned} \tag{2.14}$$

where in the final equality we have cancelled $(n_q-1)!$ with the same term in the binomial coefficient, used $\eta_q^l = \mathrm{res}_q \left( (z-q)^l \omega \right)$ and evaluated an integral over $V_q$.

## 2.2 The Equations of Motion

Our analysis is entirely classical and so the standard action's equations of motion are easily found from the variation of $S$:

$$\delta S_{\mathrm{4dCS}}(A) = \frac{1}{2\pi\hbar} \int_{\Sigma \times \mathbb{C}P^1} \omega \wedge \langle 2F(A), \delta A \rangle - \frac{1}{2\pi\hbar} \int_{\Sigma \times \mathbb{C}P^1} \omega \wedge \bar\partial \langle A, \delta A \rangle . \tag{2.15}$$

The equations of motion for $A$ follow from demanding the variation vanish.

The second term in (2.15) vanishes away from poles in $\omega$ and so the "bulk equations of motion" are found by setting the first term to zero:

$$\omega \wedge F(A) = 0 . \tag{2.16}$$

This is satisfied everywhere in $\Sigma \times \mathbb{C}P^1$.

The second term in (2.15) collapses to a sum over the poles of $\omega$ if we use (2.12) and so we call this term the "boundary variation", and the resulting equations "boundary conditions":

$$I_{\mathrm{boundary}}(A, \delta A) = \frac{1}{2\pi\hbar} \int_{\Sigma \times \mathbb{C}P^1} \omega \wedge \bar\partial \langle A, \delta A \rangle = 0 . \tag{2.17}$$

The solutions to (2.17) are a set of boundary conditions which $A$ satisfies at the poles of $\omega$, thus this equation plays a role similar to that of a boundary equation of motion. Upon imposing these boundary conditions on $A$ we introduce a set of two dimensional defects which sit at the poles of $\omega$ and span $\Sigma$; we refer to these defects as type B defects. For this reason we call equation (2.17) the defect equations of motion.

We can write down the defect equations of motion explicitly by substituting $\xi = \langle A, \delta A \rangle$ into (2.12). Firstly, we note that we require the contributions to (2.12) at each pole to vanish separately. Secondly, since we require poles on $\omega$ to be of order at most two, we can truncate the sum over $l$ in (2.12) to $l \leq 1$. Combining these, we end up with the following defect equations of motion:

$$\left( \eta_q^0 + \eta_q^1 \partial_z \right) \int_{\Sigma_q} \langle A, \delta A \rangle = 0 , \tag{2.18}$$

(for a simple pole $\eta_q^1 = 0$).

Type B defects are defined by boundary conditions which solve this equation.

Note, in the following we demand that $\langle \omega \wedge A, \delta A \rangle$ is regular. This condition will be important in the following section.

## 2.3 Boundary Conditions and Type B Defects

The 4d CS action is independent of any metric on $\Sigma$, but the two-dimensional action to which it is equivalent is either Euclidean or Lorentzian. The nature of the action is determined by the nature of the boundary conditions we impose. For simplicity, we will just discuss the Lorentzian case and will take real coordinates $x^\pm$ on $\Sigma$ which will eventually be light-cone coordinates for the Lorentzian sigma model. The Euclidean case is easily achieved by substituting complex coordinates $(w, \bar{w})$ for $(x^+, x^-)$.

In this section we introduce three classes of 'Type B' defects first given in [12]. Type B defects are solutions of (2.18) and are associated to poles in $\omega$. The first two of these classes (which we will call chiral and anti-chiral Dirichlet) are associated to first order poles in $\omega$, while the third class (which we simply call Dirichlet) is associated to a second order pole. We note that this list is not exhaustive, others are discussed in [10, 25].

Before stating the chiral, anti-chiral and Dirichlet boundary conditions we first emphasise that the gauge transformation:

$$A_\Sigma \longrightarrow A_\Sigma^u = u(d_\Sigma + A_\Sigma)u^{-1}, \tag{2.19}$$

must preserve the boundary conditions. Thus the boundary conditions also define a set of conditions on the group element $u : \Sigma \times \mathbb{C}P^1 \to G_\mathbb{C}$ which will be useful when discussing the gauge invariance of the standard action (2.2).

**Chiral Boundary Conditions:** At a simple pole $q$ where $\eta_q^1 = 0$ in (2.18) one defines the chiral boundary condition by the solution:

$$A_- = 0, \tag{2.20}$$

which also implies $\delta A_- = 0$. This boundary condition is only preserved by gauge transformations which satisfy:

$$\partial_- u = 0. \tag{2.21}$$

One uses the nomenclature 'chiral' because $A_+$ gives a chiral Kac-Moody current on the defect, as will be shown later. Note, simple poles do not contribute to the (2.11) meaning there is no issue of irregularity.

**Anti-Chiral Boundary Conditions:** Similarly, the anti-chiral boundary condition is also defined at a simple pole and is the solution of (2.18) (where again $\eta_q^1 = 0$):

$$A_+ = 0, \qquad \partial_+ u = 0, \tag{2.22}$$

where the second condition follows from the requirement the boundary condition is preserved by gauge transformations. As with the chiral case one finds $A_-$ gives anti-chiral Kac-Moody currents on the defect.

**Dirichlet Boundary Conditions:** The Dirichlet boundary conditions are defined at double poles of $\omega$ and is the following solution of (2.18):

$$A_\pm = 0, \qquad \partial_\pm u = 0, \tag{2.23}$$

which implies $\delta A_{\pm} = 0$ meaning $\partial_z \langle A, \delta A \rangle = 0$. The singular contribution to the action from a double pole (2.11) is $\partial \psi$ where:

$$\psi = -\frac{\eta_q^1}{z - q} \left( \langle A_\Sigma, \bar{\partial} A_\Sigma \rangle + \langle \bar{A}, d_\Sigma A_\Sigma \rangle + \langle A_\Sigma, d_\Sigma \bar{A} \rangle + 2 \langle \bar{A}, A_\Sigma^2 \rangle \right) . \tag{2.24}$$

This is in fact regular if Dirichlet boundary conditions hold at the pole $q$ meaning the total derivative $\partial \psi$ can be sent to zero.

## 2.4 Gauge Invariance

We have already discussed the unusual gauge invariance of the four-dimensional action; we are now in a position to discuss the physical gauge transformations. The physical gauge transformations are given by:

$$A \longrightarrow A^u = u(A + d)u^{-1}, \tag{2.25}$$

where $u \in G_\mathbb{C}$. Under such gauge transformations, the action (2.2), transforms as:

$$S_{\text{4dCS}}(A) \longrightarrow S_{\text{4dCS}}(A) + \frac{1}{2\pi\hbar} \int_{\Sigma \times \mathbb{C}P^1} \omega \wedge \bar{\partial} \langle u^{-1} du, A \rangle + \frac{1}{6\pi\hbar} \int_{\Sigma \times \mathbb{C}P^1} \omega \wedge \langle u^{-1} du, (u^{-1} du)^2 \rangle . \tag{2.26}$$

Note, we have sent a totaly derivative of $d_\Sigma$ to zero by requiring $A$ and $u$ die off to zero at infinity in $\Sigma$. In the following we denote the second term on the left hand side by $\delta S_1$ and the third by $\delta S_2$. These two terms vanish separately and we consider $\delta S_1$ first.

Using (2.12) with $\xi = \langle u^{-1} du, A \rangle$, and noting that each pole is at most second order, we find:

$$\delta S_1 = \sum_{q \in P} \delta S_q , \quad \delta S_q = \left( \eta_q^0 + \eta_q^1 \partial_z \right) \int_{\Sigma_q} \langle u^{-1} du, A \rangle = 0 . \tag{2.27}$$

where $\Sigma_q = \Sigma \times (q, \bar{q})$. We now consider the three boundary conditions in the previous section and show that the conditions on $u$ ensure $\delta S_q$ vanishes in each case.

**Chiral boundary conditions:** We take $\omega$ to have a simple pole at $z = q$ (where $\eta_q^1 = 0$), at which we impose the chiral boundary condition where $A_- = 0$ reducing equation (2.27) to:

$$\delta S_q = \eta_q^0 \int_{\Sigma_q} \langle u^{-1} \partial_- u, A_+ \rangle \, dx^- \wedge dx^+ = 0 , \tag{2.28}$$

where the final equality holds upon imposing the constraint $\partial_- u = 0$. Hence any contribution due to a first order pole in the second term of equation (2.26) can be made to vanish upon imposing chiral boundary conditions.

**Anti-chiral boundary conditions:** We take $\omega$ to have a simple pole at $z = q$, at which we impose the anti-chiral boundary condition where $A_+ = 0$. After imposing this, the term $\delta S_q$ in equation (2.27) vanishes upon imposing the constraint $\partial_+ u = 0$. Hence, any contribution due to a first order pole in the second term of equation (2.26) can be made to vanish upon imposing anti-chiral boundary conditions.

**Dirichlet boundary conditions:** Finally, we take $\omega$ to have a double at $z = q$, at which we impose the Dirichlet boundary conditions, hence (2.27) is:

$$\delta S_q = \int_{\Sigma_q} (\eta_q^0 + \eta_q^1 \partial_z) \left\langle u^{-1} du, A \right\rangle = 0 \,. \tag{2.29}$$

The condition $A_\pm = 0$ means the first term in equation (2.29) vanishes. This leaves us with:

$$\delta S_q = \int_{\Sigma_q} \eta_q^1 \partial_z \left\langle u^{-1} \partial_j u, A_k \right\rangle dx^j \wedge dx^k \,, \tag{2.30}$$

for $j, k = \pm$. Upon imposing $\partial_\pm u = 0$ along with our constraint on $A_\pm$ we find this term also vanishes. Hence any contribution due to a second order pole vanishes when we impose a Dirichlet boundary condition.

**The Wess-Zumino Term:** The final step in proving gauge invariance is to show the Wess-Zumino term:

$$\delta S_2 \equiv \int_{\Sigma \times \mathbb{C}P^1} \omega \wedge \left\langle u^{-1} du, (u^{-1} du)^2 \right\rangle \,, \tag{2.31}$$

vanishes. If we take the exterior derivative of the Wess-Zumino three form we find it is closed:

$$d \left\langle u^{-1} du, (u^{-1} du)^2 \right\rangle = - \left\langle (u^{-1} du)^2, (u^{-1} du)^2 \right\rangle = 0 \,, \tag{2.32}$$

because of cyclic symmetry of the trace. Since the three form is closed, it is natural to ask whether it is exact. We can answer this by calculating $H_{\mathrm{dR}}^3(\Sigma \times \mathbb{C}P^1)$, the third de Rham cohomology of $\Sigma \times \mathbb{C}P^1$, which is clearly dependent upon the choice of $\Sigma$. In the following sections we fix $\Sigma = \mathbb{R}^2$. In appendix B, we find $H_{\mathrm{dR}}^3(\mathbb{R}^2 \times \mathbb{C}P^1) = 0$, using the Künneth theorem and the cohomologies of $\mathbb{R}^2$ and $\mathbb{C}P^1$. As a consequence, the Wess-Zumino three form is exact on $\mathbb{R}^2 \times \mathbb{C}P^1$. If we take the three form to be the exterior derivative of $E(u)$ and integrate by parts then equation (2.31) becomes:

$$\delta S_2 = \int_{\mathbb{R}^2 \times \mathbb{C}P^1} \bar{\partial} \omega \wedge E(u) \,, \tag{2.33}$$

where we have sent a total derivative to zero by requiring $u$ die off at infinity in $\mathbb{R}^2$.

This integral is of the form (A.6) where $\xi = \xi_{+-} dx^+ \wedge dx^-$ is the $dx^+ \wedge dx^-$ component of $E(u)$. This component must take the form $\xi_{+-} = f(\zeta)_{ab} \partial_+ \zeta^a \partial_- \zeta^b$ where $\zeta$ are coordinates on the group $G$. However, we required $\partial_i u = \partial_i \zeta^a \partial_a u = 0$ at a pole of $\omega$ for either $i = +$ or $i = -$ or both. Hence $\xi_{+-}$ is zero at a pole of $\omega$. From (2.12), the integral (2.33) is given by a sum of integrals evaluated at the poles of $\omega$, and hence is zero and the four-dimensional Chern-Simons theory is gauge invariant on $\mathbb{R}^2 \times \mathbb{C}P^1$ for the boundary conditions we have discussed.

# 3 Integrable Sigma Models on Type B Defects

In this section we review [25] whose techniques will be used in the subsequent sections to derive the unified gauged sigma model. We will refer to the work of [25] as the DLMV construction. In [25], by solving the bulk and defect equations of motion and imposing suitable gauge choices, the authors reduce the four-dimensional Chern-Simons action (2.2) to the two dimensional unified sigma model action (3.33). They start by following Costello and Yamazaki by asserting the existence of a certain class of group elements $\{\hat{g}\}$ in which $A_{\bar{z}} = \hat{g} \partial_{\bar{z}} \hat{g}^{-1}$.

The next step is to show that one can work in the so-called 'archipelago gauge', expressed as a set of conditions on $\hat{g}$ called the 'archipelago' conditions. This step requires the gauge field satisfy $\omega F_{\bar{z}\pm} = 0$,

which follows from the bulk equations of motion. Finally, one can show that $A$ is gauge equivalent to a field configuration $\mathcal{L}$ which satisfies the conditions required of a Lax connection. The two dimensional unified sigma model action (3.33) is expressed in term of $\mathcal{L}$ and is completely determined (up to gauge symmetry) by the values of $\hat{g}$ (and its $\partial_{\pm}$ derivatives) at the poles of $\omega$.

## 3.1 The class $\{\hat{g}\}$ in the DLMV Construction

In [12] Costello and Yamazaki proved the existence of a class of group elements, $\hat{g}$'s, such that:

$$A_{\bar{z}}(x^{\pm}, z, \bar{z}) = \hat{g}\partial_{\bar{z}}\hat{g}^{-1}. \tag{3.1}$$

where $\hat{g} : \Sigma \times \mathbb{C}P^1 \to G_{\mathbb{C}}$. This is analogous to the construction of the Wess-Zumino-Witten model in [23] but here $\hat{g}$ is not found explicitly as a path ordered exponential. The equation (3.1) has a right acting symmetry transformation, which we call the 'right redundancy',

$$\hat{g} \longrightarrow \hat{g}' = \hat{g}k_g. \tag{3.2}$$

For $\hat{g}$ and $\hat{g}'$ to give the same field $A_{\bar{z}}$, $k_g$ must be holomorphic. However, any holomorphic function on $\mathbb{C}P^1$ is constant since $\mathbb{C}P^1$ is compact, thus $k_g$ is a function of $x^{\pm}$ only.

We will fix this right-redundancy by choosing a canonical group element $\hat{\sigma}_{\infty}$ in the class $\{\hat{g}\}$ to be the element which satisfies $\hat{\sigma}_{\infty}|_{(\infty,\infty)} = 1$, where '$|_{(q,\bar{q})}$' indicates the evaluation of a function at $\boldsymbol{z} = (q, \bar{q})$. The canonical element $\hat{\sigma}_{\infty}$ can be found from any element $\hat{g} \in \{\hat{g}\}$ via a right redundancy transformation (3.2) by $k_g = \hat{g}^{-1}|_{(\infty,\infty)}$. Under this transformation we find:

$$\hat{\sigma}_{\infty}(x^{\pm}, z, \bar{z}) = \hat{g}(x^{\pm}, z, \bar{z}) \cdot \hat{g}(x^{\pm}, \infty, \infty)^{-1}, \tag{3.3}$$

and clearly $\hat{\sigma}_{\infty}|_{(\infty,\infty)} = 1$. For the sake of brevity and clarity in the following we use $\hat{g}$ and assume it is the identity at infinity since one can always make this choice.

## 3.2 The Archipelago Conditions

In this section we correct a minor error in [25] and prove that there exists a gauge (called the 'archipelago' gauge) in which a group element $\tilde{g} \in \{\hat{g}\}$ satisfies the 'archipelago' conditions of [25]. These conditions are the following.

For each pole $q$ in $P$ we define a distance $R_q$. For each pole in $P^{\text{fin}}$, let $U_q$ be a disc of radius $R_q$ such that for a finite pole $|z - q| < R_q$; for the pole at infinity, let $U_{\infty}$ be the region $|1/z| < R_{\infty}$. We require the that the radii $R_q$ be chosen to ensure the these discs are disjoint. Using these discs we define the archipelago conditions by:

(i) $\tilde{g} = 1$ outside the disjoint union $\Sigma \times \sqcup_{q \in P} U_q$;

(ii) Within each $\Sigma \times U_q$ we require that $\tilde{g}$ depends only upon $x^{\pm}$ and the radial coordinate $r_q$ of the disc $U_q$. We choose the notation $\tilde{g}_q$ to indicate that $\tilde{g}$ is in the disc $U_q$, this condition means that $\tilde{g}_q$ is rotationally invariant;

(iii) There is an open disc $V_q \subset U_q$ centred on $q$ for every $q \in P$ such that in this disc $\tilde{g}_q$ depends upon $x^+$ and $x^-$ only. We denote $\tilde{g}_q$ in this region by $g_q = \tilde{g}|_{\Sigma \times V_q}$.

This is expressed in terms of $\tilde{g}$, but since $A_{\bar{z}}$ is determined by $\tilde{g}$, these are a partial gauge choice on $A$. The first condition sets $A_{\bar{z}} = 0$ outside $\Sigma \times \sqcup_{q \in P} U_q$. The second condition ensures that $A_{\bar{z}}$ in rotationally invariant in $U_q$. The third condition sets $A_{\bar{z}} = 0$ inside $\Sigma \times \sqcup_{q \in P} V_q$.

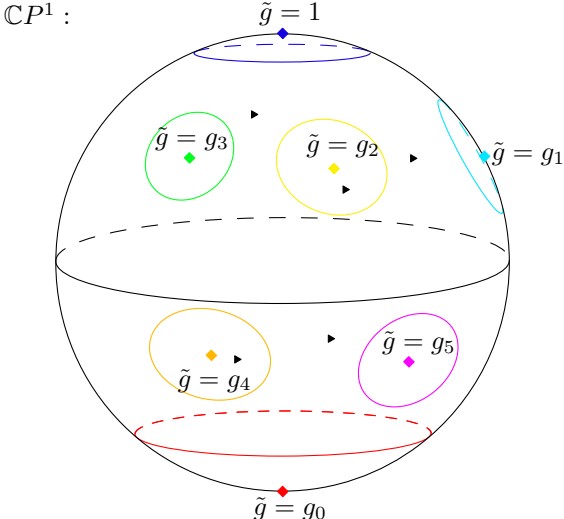

Figure 1: An illustration of the archipelago conditions for an $\omega$ with seven poles and five zeros. The diamonds represent the poles of $\omega$ with the enclosing circles illustrating the discs $U_q$. Each $\tilde{g} = g_i$ denotes the value of $\tilde{g}$ at the associated pole of $\omega$. The five black triangles represent the zeros of $\omega$ at which $A$ can have poles.

We now prove the existence of the archipelago gauge which we do in two steps. The first is, given a group element $\hat{g}$, to construct a group element $\tilde{g}$ that satisfies the archipelago conditions; the second is to show that the gauge transformation $u = \tilde{g}\hat{g}^{-1}$, which puts A into the archipelago gauge, preserves the boundary conditions on $A$.

In [25] the authors proposed a construction of $\tilde{g}$ which satisfies the archipelago conditions, but their construction is not quite right because it involves expressing $\tilde{g}$ as an exponential of a Lie algebra element. Although $\tilde{g}$ is in the identity component of $G_{\mathbb{C}}$ by the first archipelago condition it is not the case that $\tilde{g}$ can be constructed as an exponential everywhere in the identity component. For example, if we take $G_{\mathbb{C}} = SL(2, \mathbb{C})$ then the group element:

$$\begin{pmatrix} -1 & 1 \\ 0 & -1 \end{pmatrix} , \tag{3.4}$$

is in the identity component of $SL(2, \mathbb{C})$ but cannot be written as an exponential of an element of the Lie algebra $\mathbf{sl}(2, \mathbb{C})$. It is for this reason that the treatment presented below is slightly different to that presented in [25].

This minor issue is easily solved by the following argument. Let $\hat{g}$ be the original group element and $\tilde{g}$ the group element in archipelago gauge.

We consider each pole $q$ in turn. We first choose the value of $\tilde{g}$ at the pole to be the same as that of $\hat{g}$, so that

$$g_q = \hat{g}|_{(q,\bar{q})} . \tag{3.5}$$

Since $\hat{g}|_{(\infty,\infty)} = 1$ and $\hat{g}$ varies smoothly over $M$, $\hat{g}$ must be in the identity component of $G$ everywhere on $M$. This means, for each disc $U_q$, we can choose a path in $G$ between 1 and $g_q$. We parametrise this path

as $\tilde{g}_q(r_q)$ where $\tilde{g}_q(R_q) = 1$ and $\tilde{g}_q(r_q) = g_q$ when $r_q$ is in $V_q$. We then define $\tilde{g}$ to be

$$\tilde{g}(z, \bar{z}, x^+, x^-) = \begin{cases} 1 & z \in \mathbb{C}P^1 \setminus \sqcup_{q \in P} U_q \\ \tilde{g}_q(r_q, x^+, x^-) & z \in U_q \setminus V_q \\ g_q(x^+, x^-) & z \in V_q \end{cases} \tag{3.6}$$

As constructed, $\tilde{g}$ in (3.6) satisfies the archipelago conditions.

We now show that the gauge transformation $u = \tilde{g}\hat{g}^{-1}$ from $\hat{g}$ to $\tilde{g} = u\hat{g}$ satisfies the boundary condition (2.21), (2.22) and (2.23), as appropriate. By equations (3.6) and (3.5) it is clear $u = 1$ at a pole $q \in P$, thus $\partial_\pm u = 0$. Hence the boundary conditions are satisfied and the archipelago gauge accessible.

## 3.3 Lax connection

In this subsection we introduce the notion of a Lax connection $\mathcal{L}$ for an integrable model. We follow [25] and prove that $A$ is gauge equivalent to a Lax connection $\mathcal{L}$, using the equations of motion and Wilson lines of four-dimensional Chern-Simons theory. We follow this with a discussion of the gauge transformations of $\mathcal{L}$ induced by those $A$, from which it follows that there exists an equivalence class of Lax connections. This is as one would expect since a sigma model should not have a preferred Lax connection. We conclude this section by giving the generic form of the Lax connection as in [39].

A one-form $\mathcal{L}$ on a two-dimensional space $\Sigma$ with coordinates $(x^1, x^2)$, with components given as functions of the degrees of freedom of an integrable model, is called a Lax connection if it satisfies the following properties: [40]:

1. The equation $d_\Sigma \mathcal{L} + \mathcal{L} \wedge \mathcal{L} = 0$ gives the equations of motion for the integrable model.

It is called a Lax connection with spectral parameter if, in addition, it satisfies

2. $\mathcal{L}$ has a meromorphic dependence upon on a complex parameter $z$, called the spectral parameter,

3. One can obtain an infinite set of Poisson-commuting charges from the expansion in $z$ of the monodromy matrix. The monodromy matrix is the path ordered exponential of the line integral of $\mathcal{L}$.

We can construct such a one-form $\mathcal{L}$ from a gauge field $A$ by taking the gauge transformation by the group element $\tilde{g}$ (and an additional special gauge transformation to remove the $z$ component):

$$\mathcal{L} = \tilde{g}^{-1}d\tilde{g} + \tilde{g}^{-1}A\tilde{g} + \chi_z dz . \tag{3.7}$$

By construction, $\mathcal{L}_z = \mathcal{L}_{\bar{z}} = 0$ which is ensured by setting $\chi_z = -\tilde{g}^{-1}\partial_z\tilde{g}$. The equations of motion (2.16) for $A$ imply that

$$\omega \wedge \bar{\partial}\mathcal{L} = 0 , \qquad d_\Sigma\mathcal{L} + \mathcal{L} \wedge \mathcal{L} = 0 . \tag{3.8}$$

The first of these equations means $\mathcal{L}$ has a meromorphic dependence upon $z$ and thus satisfy the second of property of a Lax connection. As was discussed above, meromorphic one-forms are of the form (3.20) for an $\omega$ with a pole at infinity. The second means $\mathcal{L}$ is flat in the plane $\Sigma$, and was shown to necessarily give the sigma models equations of motion in [39].

The final property of the Lax connection follows from the Wilson line operators in four-dimensional Chern-Simons theory. Here we present the case where $\Sigma = S^1 \times \mathbb{R}$ for illustrative purposes which can be found in [40]. The generic case follows from [24] in which Vicedo found that the four-dimensional Chern-Simons Poisson algebra (in an appropriate gauge) is that of a Lax connection. The conservation and Poisson commutativity of the infinite stack of charges then follows from the standard argument found in [41, 42].

The monodromy matrix of $\mathcal{L}$ is:

$$U(z,t) = P\exp\left(\int_0^{2\pi} \mathcal{L}_\theta d\theta\right) = \tilde{g}^{-1}P\exp\left(\int_0^{2\pi} A_\theta d\theta\right)\tilde{g}\,, \qquad (3.9)$$

Following the standard argument, we take the trace of both matrices and find:

$$W(z,t) = \text{Tr}P\exp\left(\int_0^{2\pi} \mathcal{L}_\theta d\theta\right) = \text{Tr}P\exp\left(\int_0^{2\pi} A_\theta d\theta\right)\,, \qquad (3.10)$$

where the right-hand side is a gauge invariant observable in four-dimensional Chern-Simons, implying the trace of the monodromy matrix is an observable. By taking the time derivative of $W(z,t)$ we find:

$$\partial_t W(z,t) = \text{Tr}\left([U(z,t), \mathcal{L}_\theta]\right) = 0\,, \qquad (3.11)$$

and thus that $W(z)$ is independent of their position along the length of the cylinder. By Taylor expanding $W(z)$ in $z$ it follows from (3.11) that the coefficient of each power is conserved in time. This set of coefficients is the infinite stack of charges associated to $\mathcal{L}$, they are observables since $W_A(z)$ is.

We now turn to a discussion of the gauge symmetry of $\mathcal{L}$. At the beginning of this section we discussed the right redundancy (3.2) of the class of group elements $\{\hat{g}\}$. The right redundancy amongst $\{\hat{g}\}$ left $A_{\bar{z}}$ invariant meaning every element must give the same field configuration which is completely determined in terms of $A_{\bar{z}}$ by the equations of motion. By performing a right redundancy transformation on $\tilde{g}$ in (3.7), and using the fact that $A$ is invariant, we find the induced gauge transformation of $\mathcal{L}$:

$$\begin{aligned}
\mathcal{L} \longrightarrow \mathcal{L}^h &= (\tilde{g}h)^{-1}A(\tilde{g}h) + (\tilde{g}h)^{-1}d(\tilde{g}h) \\
&= h^{-1}(\tilde{g}^{-1}A\tilde{g} + \tilde{g}^{-1}d\tilde{g})h + h^{-1}dh \\
&= h^{-1}\mathcal{L}h + h^{-1}dh\,,
\end{aligned} \qquad (3.12)$$

where we have used $\partial_z h = 0$. Hence, $A$ is left invariant under the combined transformations:

$$\tilde{g} \longrightarrow \tilde{g}h, \quad \mathcal{L} \longrightarrow \mathcal{L}^h = h^{-1}\mathcal{L}h + h^{-1}dh\,. \qquad (3.13)$$

The invariance of $A$ under the right redundancy is significant as it means that a field configuration $A$ is associated to a class of gauge equivalent Lax connections (via the right redundancy). This means that there is no preferred Lax connection, as one would expect given an integrable sigma model. This fact can be concretely proven by noting that any sigma model found by substituting a field configuration $A$ is left invariant by the right redundancy because $A$ is. This was shown explicitly in [25].

As our final remark we discuss the transformation of $\mathcal{L}$ induced by gauge transformations of $A$, equation (2.25). Consider the inverse gauge transformation of (3.7) by $\tilde{g}$:

$$A = \tilde{g}d\tilde{g}^{-1} + \tilde{g}\mathcal{L}\tilde{g}^{-1}\,, \qquad (3.14)$$

hence under the gauge transformation (2.25) we find:

$$A \longrightarrow udu^{-1} + uAu^{-1} = (u\tilde{g})d(u\tilde{g})^{-1} + (u\tilde{g})\mathcal{L}(u\tilde{g})^{-1}\,, \qquad (3.15)$$

and thus that a gauge transformation is equivalent to $\tilde{g} \to u\tilde{g}$ for an arbitrary element $\tilde{g} \in \{\tilde{g}\}$. To remove the right redundancy in (3.1) we work with the canonical element (3.3) and use $\hat{\sigma}_\infty \to u\hat{\sigma}_\infty$, thus under a gauge transformation the canonical element transforms as:

$$\hat{\sigma}_\infty = \tilde{g}\cdot(\tilde{g}^{-1}|_{(\infty,\infty)}) \longrightarrow u\tilde{g}\cdot(\tilde{g}^{-1}|_{(\infty,\infty)})u_\infty^{-1} = u\hat{\sigma}_\infty u_\infty^{-1}\,, \qquad (3.16)$$

where $u_\infty = u|_{(\infty,\infty)}$ appears because we are fixing the right redundancy at infinity. Clearly $\mathcal{L}$ in (5.4) is only well defined when the right redundancy is fixed which we do with canonical elements (3.3). Thus, once the right redundancy is fixed it follows that the transformations of $\mathcal{L}$ induced by gauge transformations of $A$ are of form:

$$\mathcal{L} \longrightarrow (u_\infty \tilde{g}^{-1} u^{-1}) d(u\tilde{g}u_\infty^{-1}) + (u_\infty \tilde{g}^{-1} u^{-1}) A^u (u\tilde{g}u_\infty^{-1}) = u_\infty du_\infty^{-1} + u_\infty \mathcal{L} u_\infty^{-1}, \tag{3.17}$$

where we have assumed $\tilde{g}$ is an canonical element.

## 3.4 Type A Defects and the Equations of Motion for $\omega$ with Zeros

When deriving the equations of motion we sent the total derivative $d(\omega \wedge A \wedge \delta A)$ to zero implicitly assuming that $\omega \wedge A \wedge \delta A$ contains no poles. Thus at a zero $\zeta$ of $\omega$ which is of order $m_\zeta$ it follows that $A \wedge \delta A$ can have a pole whose order is at most $m_\zeta$. Due to the wedge product in $A \wedge \delta A$ the order of this pole is the sum of the orders of the poles of $A_+$ and $A_-$.

The poles of $A_\pm$ are two dimensional defects and are called type A defects. A classification of these defects was given in [12]; we rephrase this classification as the following regularity conditions on $A$ at the zeros of $\omega$.

We define a type A defect of type $(k_\zeta^+, k_\zeta^-)$ at a zero $\zeta$ by the conditions:

- $(z - \zeta)^{k_\zeta^+} A_+$ is regular;

- $(z - \zeta)^{k_\zeta^-} A_-$ is regular.

These boundary conditions must be preserved by gauge transformations, which occurs if the gauge transformations are regular at the zeros of $\omega$.

In the following we discuss the solution to $\omega \wedge \bar{\partial}\mathcal{L} = 0$. For $\omega$ with first order zeros and at most double poles the solution was given in [25] while the generic solution can be found in [39]. The inclusion of zeros in $\omega$ is of significance as it allows for poles in the gauge field. Following [25] and [39] our Lax connections have a partial fraction expansion of the form:

$$\mathcal{L}_\pm = \mathcal{L}_\pm^c(x^+, x^-) + \left( \sum_{\zeta \in Z \setminus \{\infty\}} \sum_{l=0}^{k_\zeta^\pm - 1} \frac{\mathcal{L}_\pm^{\zeta,l}(x^+, x^-)}{(z-\zeta)^{l+1}} \right) + \sum_{l=0}^{k_\infty^\pm - 1} \mathcal{L}_\pm^{\infty,l}(x^+, x^-) z^{l+1}. \tag{3.18}$$

where $\mathcal{L}_\pm^c, \mathcal{L}_\pm^{\infty,l}, \mathcal{L}_\pm^{\zeta,l} : \Sigma \to \mathbf{g}_\mathbb{C}$ while $\mathcal{L}_\pm^l = \mathrm{res}_\zeta((z-\zeta)^l \mathcal{L}_\pm)$ and $\mathcal{L}_\pm^{\infty,l} = \mathrm{res}_\infty(\mathcal{L}_\pm/z^l)$. The positive integer $k_\zeta^+$ (resp. $k_\zeta^-$) is the highest order of the pole of $\mathcal{L}_+$ (resp. $\mathcal{L}_-$) at $\zeta$.

Rather than simply stating (3.18), we feel obliged to explain why $\mathcal{L}$ is of this form. The one-form $\omega$ has a finite set of zeros away from which the equality in:

$$\omega \wedge \bar{\partial}\mathcal{L}_\pm = 0, \tag{3.19}$$

holds only if $\bar{\partial}\mathcal{L}_\pm = 0$ meaning $\mathcal{L}_\pm$ is not a function of $\bar{z}$. However, at the zeros of $\omega$ the equality in (3.19) holds even when $\bar{\partial}\mathcal{L}_\pm \neq 0$. Thus, $\bar{\partial}\mathcal{L}_\pm$ has finite support meaning $\mathcal{L}_\pm$ is meromorphic in $z$ with poles the zeros of $\omega$, this is because $\bar{\partial}$ derivatives of poles are delta functions. We note, the equality in (3.19) only holds at a zero of $\omega$ if the pole of $\mathcal{L}_\pm$ is of the same order or less than the multiplicity of the zero. On $\mathbb{C}P^1$ meromorphic functions are ratios of two polynomials in $z$ leading to the partial fraction expansion (3.18). The polynomial terms of (3.18) follow from the assumption that $\omega$ has a zero at $z = \infty$, these terms are clearly poles by the inversion $z \to 1/z$.

When $\varphi(z)$ has a pole at infinity (3.18) reduces to:

$$\mathcal{L}_\pm = \mathcal{L}_\pm^c(x^+, x^-) + \sum_{\zeta \in Z} \sum_{l=0}^{k_\zeta^\pm - 1} \frac{\mathcal{L}_\pm^{\zeta,l}(x^+, x^-)}{(z-\zeta)^{l+1}} \,, \tag{3.20}$$

where $\mathcal{L}_\pm^c = \lim_{z \to \infty} \mathcal{L}_\pm$. In the following $\mathcal{L}_\pm^{\zeta,l}$ and $\mathcal{L}_\pm^c$ are fixed by the boundary conditions on $A$ at the poles of $\omega$. This means $\mathcal{L}_\pm^{\zeta,l}$ and $\mathcal{L}_\pm^c$ are functions of $\tilde{g}$ evaluated at the poles of $\omega$, $\{\tilde{g}_q\}$, which are the fields of our sigma models once a field configuration $A$ is substituted into the action.

## 3.5 The Unified Sigma Model Action

In this section we rewrite the four-dimensional Chern-Simons action in terms of $\tilde{g}$ and $\mathcal{L}$ and use the archipelago conditions to reduce the four-dimensional action to the two-dimensional unified sigma model of [25] defined on the defects at the poles of $\omega$. To do this we substitute:

$$A = \tilde{g} d\tilde{g}^{-1} + \tilde{g}\mathcal{L}\tilde{g}^{-1} \,, \tag{3.21}$$

into the four-dimensional Chern-Simons action by using:

$$CS(\hat{A} + A') = CS(\hat{A}) + CS(A') - d\left\langle \hat{A}, A' \right\rangle + 2\left\langle F(\hat{A}), A' \right\rangle + 2\left\langle \hat{A}, A' \wedge A' \right\rangle \,, \tag{3.22}$$

where we take $A = \hat{A} + A'$ such that:

$$\hat{A} = \tilde{g} d\tilde{g}^{-1} \,, \qquad A' = \tilde{g}\mathcal{L}\tilde{g}^{-1} \,. \tag{3.23}$$

Straight away we can see the third term of equation (3.22) vanishes as $F(\hat{A}) = 0$, while the first term is:

$$CS(\hat{A}) = \frac{1}{3}\left\langle \tilde{g}^{-1} d\tilde{g}, \tilde{g}^{-1} d\tilde{g} \wedge \tilde{g}^{-1} d\tilde{g} \right\rangle \,. \tag{3.24}$$

In $CS(A')$ the second term $A' \wedge A' \wedge A'$ vanishes as $\mathcal{L}$ is a one form with non-zero $\Sigma$ components only. Hence we are only concerned with the kinetic term, which is:

$$CS(A') = \langle A', dA' \rangle = \left\langle \tilde{g}\mathcal{L}\tilde{g}^{-1}, d\tilde{g} \wedge \mathcal{L}\tilde{g}^{-1} \right\rangle + \left\langle \tilde{g}\mathcal{L}\tilde{g}^{-1}, g d\mathcal{L}\tilde{g}^{-1} \right\rangle - \left\langle \tilde{g}\mathcal{L}\tilde{g}^{-1}, \tilde{g}\mathcal{L} \wedge d\tilde{g}^{-1} \right\rangle \,, \tag{3.25}$$

which we simplify by taking $d\tilde{g} = -\tilde{g}d\tilde{g}^{-1}\tilde{g}$ in the first term, as well as by inserting $\tilde{g}^{-1}\tilde{g}$ between $\tilde{g}\mathcal{L}$ and $d\tilde{g}^{-1}$. Having done this we find:

$$CS(A') = -\left\langle \tilde{g}\mathcal{L}\tilde{g}^{-1}, \tilde{g}d\tilde{g}^{-1} \wedge \tilde{g}\mathcal{L}\tilde{g}^{-1} \right\rangle + \langle \mathcal{L}, d\mathcal{L} \rangle - \left\langle \tilde{g}\mathcal{L}\tilde{g}^{-1}, \tilde{g}\mathcal{L}\tilde{g}^{-1} \wedge \tilde{g}\mathcal{L}\tilde{g}^{-1} \right\rangle \,, \tag{3.26}$$

but $\tilde{g}\mathcal{L}\tilde{g}^{-1} \wedge \tilde{g}\mathcal{L}\tilde{g}^{-1} \wedge \tilde{g}\mathcal{L}\tilde{g}^{-1}$ is just $A' \wedge A' \wedge \hat{A}$, therefore:

$$CS(A') = \langle \mathcal{L}, d\mathcal{L} \rangle - 2\left\langle \hat{A}, A' \wedge A' \right\rangle \,, \tag{3.27}$$

which cancels with $2\left\langle \hat{A}, A' \wedge A' \right\rangle$ of (3.22). Hence, upon simplifying the fourth term we find:

$$CS(\hat{A} + A') = \langle \mathcal{L}, d\mathcal{L} \rangle + d\left\langle \tilde{g}^{-1} d\tilde{g}, \mathcal{L} \right\rangle + \frac{1}{3}\left\langle \tilde{g}^{-1} d\tilde{g}, \tilde{g}^{-1} d\tilde{g} \wedge \tilde{g}^{-1} d\tilde{g} \right\rangle \,, \tag{3.28}$$

where we have used $d\left\langle \hat{A}, A'\right\rangle = -d\left\langle \tilde{g}^{-1}d\tilde{g}, \mathcal{L}\right\rangle$ in equation (3.22). This leaves us with the action:

$$S_{\text{4dCS}}(A) = \frac{1}{2\pi\hbar}\int_{\Sigma\times\mathbb{C}P^1}\omega\wedge\left\langle\mathcal{L}, \bar{\partial}\mathcal{L}\right\rangle - \frac{1}{2\pi\hbar}\int_{\Sigma\times\mathbb{C}P^1}\omega\wedge\bar{\partial}\left\langle\mathcal{L}, \tilde{g}^{-1}d\tilde{g}\right\rangle$$
$$+ \frac{1}{6\pi\hbar}\int_{\Sigma\times\mathbb{C}P^1}\omega\wedge\left\langle\tilde{g}^{-1}d\tilde{g}, \tilde{g}^{-1}d\tilde{g}\wedge\tilde{g}^{-1}d\tilde{g}\right\rangle . \tag{3.29}$$

One recovers sigma models from the above action by substituting in solutions to the equations of motion, thus we take $\mathcal{L}$ to be of the form (3.20). The derivative $\bar{\partial}$ reduces the power of a pole of (3.20) at $\zeta$ by a factor of one and introduces a delta function, thus the pole of $\mathcal{L}\wedge\bar{\partial}\mathcal{L}$ at $\zeta$ is of degree $k_\zeta^+ + k_\zeta^- - 1$. However, the zero $\zeta$ of $\omega$ is of degree $m_\zeta \geqslant k_\zeta^+ + k_\zeta^-$ hence each term of $\omega\wedge\mathcal{L}\wedge\bar{\partial}\mathcal{L}$ contains a zero of at least degree one and thus vanishes. Therefore, (3.29) reduces to:

$$S_{\text{4dCS}}(A) = -\frac{1}{2\pi\hbar}\int_{\Sigma\times\mathbb{C}P^1}\omega\wedge\bar{\partial}\left\langle\mathcal{L}, \tilde{g}^{-1}d\tilde{g}\right\rangle + \frac{1}{6\pi\hbar}\int_{\Sigma\times\mathbb{C}P^1}\omega\wedge\left\langle\tilde{g}^{-1}d\tilde{g}, \tilde{g}^{-1}d\tilde{g}\wedge\tilde{g}^{-1}d\tilde{g}\right\rangle . \tag{3.30}$$

This action can be reduced further using the archipelago conditions as shown in [25]. The first term is easily calculated using (2.12) where we find:

$$-\frac{1}{2\pi\hbar}\int_{\Sigma\times\mathbb{C}P^1}\omega\wedge\bar{\partial}\left\langle\mathcal{L}, \tilde{g}^{-1}d\tilde{g}\right\rangle = -\frac{i}{\hbar}\sum_{q\in P}\int_{\Sigma_q}\left\langle\text{res}_q(\omega\wedge\mathcal{L}), g_q^{-1}dg_q\right\rangle , \tag{3.31}$$

having used the third archipelago condition to remove $g_q^{-1}dg_q$ from the residue.

A detailed reduction of the final term of (3.30) can be found in appendix C however, the summary of this calculation is the following: the first archipelago condition lets us localise the second integral of (3.30) to the regions $U_q$ in which each $\tilde{g}_q$ is rotationally invariant. Outside of $\sqcup_q U_q$ we have $\tilde{g} = 1$, meaning $\mathbb{C}P^1 \setminus \sqcup_q U_q$ does not contribute to the integral. Next, one changes coordinates to polar coordinates and performs the angular integral, from which ones finds (C.3):

$$\frac{1}{6\pi\hbar}\int_{\Sigma\times\mathbb{C}P^1}\omega\wedge\left\langle\tilde{g}^{-1}d\tilde{g}, (\tilde{g}^{-1}d\tilde{g})^2\right\rangle = \frac{i}{3\hbar}\sum_{q\in P}\text{res}_q(\omega)\int_{\Sigma\times[0,R_q]}\left\langle\tilde{g}_q^{-1}d\tilde{g}_q, (\tilde{g}_q^{-1}d\tilde{g}_q)^2\right\rangle . \tag{3.32}$$

Upon combining this together, we find the unified sigma model action:

$$S_{\text{USM}}(\mathcal{L}, \tilde{g}) \equiv S_{\text{4dCS}}(A) = -\frac{i}{\hbar}\sum_{q\in P}\int_{\Sigma_q}\left\langle\text{res}_q(\omega\wedge\mathcal{L}), g_q^{-1}dg_q\right\rangle \tag{3.33}$$
$$+ \frac{i}{3\hbar}\sum_{q\in P}\text{res}_q(\omega)\int_{\Sigma\times[0,R_q]}\left\langle\tilde{g}_q^{-1}d\tilde{g}_q, (\tilde{g}_q^{-1}d\tilde{g}_q)^2\right\rangle .$$

## 3.6 Examples of Models

The choice of 1-form $\omega$ and types of defects does not necessarily lead to an interesting sigma model, or to a Lax connection with spectral parameter. To illustrate this, we consider the cases of (a) no zeros, one double pole with Dirichlet boundary conditions (b) no zeros, two simple poles, both with chiral boundary conditions (c) no zeros, two simple poles, with chiral and anti-chiral boundary conditions (d) two zeros, two double poles with Dirichlet boundary conditions. To derive these models we use the solution to the equations of motion in the gauge $A_{\bar{z}} = 0$, equation (3.20), with an appropriately chosen $\omega$ and boundary conditions on $A$, to find $\mathcal{L}$. This is done using:

$$\mathcal{L}_\pm|_{(q,\bar{q})} = g_q^{-1}A_\pm|_{(q,\bar{q})}g_q + g_q^{-1}\partial_\pm g_q . \tag{3.34}$$

Having found $\mathcal{L}$ we calculate the unified sigma model (3.33). For simplicity's sake, we specialise to the case where $\Sigma = \mathbb{R}^2$, which we parametrise with light-cone coordinates $x^+$, and $x^-$. Note, the archipelago conditions are compatible with the boundary conditions we use in this section, thus we work in the archipelago gauge and with the group element $\tilde{g}$. Each of these examples has a pole at infinity at which we fix the right redundancy with $g_\infty = 1$.

### 3.6.1 $\omega$ with a Single Double Pole

Consider $\omega$ of the form:

$$\omega = dz\,, \tag{3.35}$$

which has a single double pole at infinity. We impose the Dirichlet boundary condition at this pole:

$$A_\pm|_{(\infty,\infty)} = 0\,. \tag{3.36}$$

Since $\omega$ has no zeros the solution (3.20) is of the form:

$$\mathcal{L} = \mathcal{L}_+^c(x^+, x^-)dx^+ + \mathcal{L}_-^c(x^+, x^-)dx^-\,. \tag{3.37}$$

If we substitute this equation into (3.34) and use $g_\infty = 1$ to fix the right redundancy it follows that the Dirichlet boundary condition implies:

$$\mathcal{L} = 0\,. \tag{3.38}$$

Similarly, the evaluation of $\mathrm{res}_\infty(\omega)$ vanishes. Hence, when we calculate the unified sigma model action (3.33) we find it vanishes.

### 3.6.2 $\omega$ with Two Simple Poles and a Common Boundary Condition

One must be careful to choose boundary conditions which completely fix the field configuration. For example, take $\omega$ to be:

$$\omega = \frac{dz}{z}\,, \tag{3.39}$$

with simple poles at both zero and infinity at which we impose chiral boundary conditions:

$$A_-|_{(0,0)} = 0\,, \qquad A_-|_{(\infty,\infty)} = 0\,. \tag{3.40}$$

As in the previous example, $\omega$ does not have any zeros meaning we again consider solutions of the form:

$$\mathcal{L} = \mathcal{L}_+^c(x^+, x^-)dx^+ + \mathcal{L}_-^c(x^+, x^-)dx^-\,. \tag{3.41}$$

Using $g_\infty = 1$ in (3.34) it follows the boundary condition at infinity implies:

$$\mathcal{L}_- = 0\,, \qquad \mathcal{L}_+ = B_+(x^+, x^-)\,, \tag{3.42}$$

where $B_+ = A_+|_{(\infty,\infty)}$. Similarly, at $z = 0$ we have $g_0 = g$ and thus by (3.34):

$$\mathcal{L}_- = g^{-1}\partial_- g = 0\,, \qquad \mathcal{L}_+ = g^{-1}\partial_+ g + g^{-1}A_+|_{(0,0)}g = B_+\,. \tag{3.43}$$

Hence, the boundary conditions (3.40) constrain $\tilde{g}$ to be a function of $x^+$ only at $z = 0$ and fail to fix $A_+$ as it is expressed everywhere in terms of the undetermined field $B_+$. Further still, this solution leads to a vanishing sigma model action when substituted into (3.33). This is because the conditions $g_\infty = 1$ and $g_0 = g(x^+)$ ensure both the kinetic and Wess-Zumino terms vanish.

### 3.6.3 $\omega$ with Two Simple Poles and Different Boundary Conditions

As shown in [12] one can recover the Wess-Zumino-Witten model for $\omega$ of the form:

$$\omega = \frac{dz}{z}\,, \tag{3.44}$$

if the appropriate boundary conditions are chosen. To be precise, we choose a pair of chiral and anti-chiral boundary conditions:

$$A_-|_{(0,0)} = 0\,, \qquad A_+|_{(\infty,\infty)} = 0\,. \tag{3.45}$$

In addition, we fix $\tilde{g}$ to be of the form:

$$\tilde{g}|_{(0,0)} = g_0 = g\,, \qquad \tilde{g}|_{(\infty,\infty)} = g_\infty = 1\,. \tag{3.46}$$

Again, $\mathcal{L}$ is of the form:

$$\mathcal{L} = \mathcal{L}_+^c(x^+, x^-)dx^+ + \mathcal{L}_-^c(x^+, x^-)dx^-\,. \tag{3.47}$$

Inserting these conditions into (3.34) we find:

$$\mathcal{L}_+ = 0\,, \qquad \mathcal{L}_- = g^{-1}\partial_- g\,. \tag{3.48}$$

This is an example of a Lax connection without a spectral parameter.

If we substitute this solution into (3.33) one finds the only non-zero contribution to the kinetic term is:

$$\text{res}_0(\omega \wedge \mathcal{L}) = g^{-1}\partial_- g\, dx^-\,, \tag{3.49}$$

while the Wess-Zumino terms vanishes at infinity since $\tilde{g}$ is the identity at $r_0$ and $r_\infty = R_\infty$. Hence we find the Wess-Zumino-Witten model:

$$S_{\text{WZW}}(g) = \frac{k}{4\pi}\int_{\Sigma_0} d^2x \left\langle g^{-1}\partial_+ g, g^{-1}\partial_- g\right\rangle + \frac{k}{12\pi}\int_{\Sigma\times[0,R_0]} \left\langle \tilde{g}^{-1}d\tilde{g}, (\tilde{g}^{-1}d\tilde{g})^2\right\rangle\,, \tag{3.50}$$

where $d^2x = dx^+ \wedge dx^-$ and $-i\hbar = 4\pi/k$.

### 3.6.4 The Principal Chiral Model with Wess-Zumino Term

For the purpose of illustrating the case where $\omega$ has zeros, we repeat the derivation of the principal chiral model with Wess-Zumino term done in [12, 25]. This should aid the clarity of subsequent sections.

We consider the four-dimensional Chern-Simons theory where $\omega$ is given by:

$$\omega = \frac{(z - \zeta_+)(z - \zeta_-)}{z^2}dz\,. \tag{3.51}$$

At the zero $z = \zeta_+$ we insert a $(1,0)$-defect such that $(z - \zeta_+)A_+$ and $A_-$ are regular, while at $z = \zeta_-$ we insert a $(0,1)$-defect such that $A_+$ and $(z - \zeta_-)A_-$ are regular. This allows a first order pole in $A_+$ at $z = \zeta_+$ and a first order pole in $A_-$ at $z = \zeta_-$. Hence, upon using (3.20) (as (3.51) has a pole at infinity) it follows our Lax connection is of the form:

$$\mathcal{L} = \left(\mathcal{L}_+^c + \frac{\mathcal{L}_+^{\zeta,0}}{z - \zeta_+}\right)dx^+ + \left(\mathcal{L}_-^c + \frac{\mathcal{L}_-^{\zeta,0}}{z - \zeta_-}\right)dx^-\,. \tag{3.52}$$

Our chosen one-form $\omega$ (3.51) has a doubles pole at both $z = 0$ and $z = \infty$, at which we impose the Dirichlet boundary conditions:

$$A|_{(0,0)} = 0\,, \qquad A|_{(\infty,\infty)} = 0\,. \tag{3.53}$$

Further, we fix $\tilde{g}$ to be the identity at infinity (fixing the right redundancy, as described above) and at $z = 0$ we denote it by $g$, thus:

$$\tilde{g}|_{(0,0)} = g_0 = g \,, \qquad \tilde{g}|_{(\infty,\infty)} = g_\infty = 1 \,. \tag{3.54}$$

By inserting this into equation (3.34) we find:

$$\mathcal{L}_\pm|_{(0,0)} = g^{-1}\partial_\pm g + g^{-1}A_\pm|_{(0,0)}g \,, \qquad \mathcal{L}_\pm|_{(\infty,\infty)} = A_\pm|_{(\infty,\infty)} \,, \tag{3.55}$$

which we use to fix $\mathcal{L}_\pm^c$ and $\mathcal{L}_\pm^{\zeta,0}$ in terms of $g$. By using the boundary condition on $A$ at $z = \infty$ the second of these two equations implies:

$$\mathcal{L}_\pm^c = 0 \,, \tag{3.56}$$

while the first equation gives:

$$\mathcal{L}_\pm^{\zeta,0} = -\zeta_\pm g^{-1}\partial_\pm g \,. \tag{3.57}$$

Hence, we find the Lax connection of the principal chiral model with Wess-Zumino term:

$$\mathcal{L} = -\frac{\zeta_+}{z - \zeta_+}g^{-1}\partial_+g\,dx^+ - \frac{\zeta_-}{z - \zeta_-}g^{-1}\partial_-g\,dx^- \,. \tag{3.58}$$

Since we work in the archipelago gauge, the action is of the form (3.33), from which we recover the sigma model action by substituting in (3.58). Both of the poles of (3.51) contribute to the kinetic term of (3.33), however the term at infinity vanishes because $g_\infty = 1$. Thus we need only evaluate $\text{res}_0(\omega \wedge \mathcal{L})$:

$$\text{res}_0(\omega \wedge \mathcal{L}) = -\zeta_+g^{-1}\partial_+g\,dx^+ - \zeta_-g^{-1}\partial_-g\,dx^- \,. \tag{3.59}$$

Similarly, the coefficient of the Wess-Zumino term at $z = 0$ is $\text{res}_0(\omega) = -(\zeta_+ + \zeta_-)$ and we needn't calculate $\text{res}_\infty(\omega)$ since the associated Wess-Zumino term vanishes as $\tilde{g}$ is the identity at $r_\infty = 0$ and $r_\infty = R_\infty$. Therefore, we find the principal chiral model with Wess-Zumino term[3]:

$$S_{\text{PMC+WZ}}(g) = i\frac{\zeta_+ - \zeta_-}{\hbar}\int_{\mathbb{R}_0^2}d^2x\,\left\langle g^{-1}\partial_+g, g^{-1}\partial_-g\right\rangle - i\frac{\zeta_+ + \zeta_-}{3\hbar}\int_{\mathbb{R}^2\times[0,R_0]}\left\langle\tilde{g}^{-1}d\tilde{g}, (\tilde{g}^{-1}d\tilde{g})^2\right\rangle \,, \tag{3.60}$$

where $\mathbb{R}_0^2 = \mathbb{R}^2 \times (0,0)$.

As a final remark, it is interesting to consider two limits of this theory. The first limit of interest is $\zeta_+ \to 0$ in which we recover the Wess-Zumino-Witten model (3.50), where we have set $i\hbar = 4\pi$ and $\zeta_- = k$. The second interesting limit is $\zeta_+ \to \zeta_-$ in which the kinetic term vanishes leaving us with a topological sigma model.

# 4  Doubled Four-Dimensional Chern-Simons

Four-dimensional Chern-Simons described integrable sigma models because the gauge field $A$ is gauge equivalent to a Lax connection. It therefore follows that a set of four-dimensional Chern-Simons theories will describe a collection of integrable models. In this section we ask whether multiple four-dimensional Chern-Simons theories can be coupled together and still describe an integrable model.

The simplest version of this theory contains two gauge fields: $A$ with the gauge group $G_\mathbb{C}$ and $B$ with the gauge group $H_\mathbb{C} \subseteq G_\mathbb{C}$. We respectively denote the Lie algebras of $G_\mathbb{C}$ and $H_\mathbb{C}$ by $\mathbf{g}_\mathbb{C}$ and $\mathbf{h}_\mathbb{C} \subseteq \mathbf{g}_\mathbb{C}$. In this section we take $\mathbf{g}_\mathbb{C}$ and $\mathbf{h}_\mathbb{C}$ to be semisimple and $\pi : \mathbf{h}_\mathbb{C} \hookrightarrow \mathbf{g}_\mathbb{C}$ to be the embedding of $\mathbf{h}_\mathbb{C}$ into $\mathbf{g}_\mathbb{C}$.

We define the bilinear form $\langle\cdot,\cdot\rangle_\mathbf{g}$ on $\mathbf{g}_\mathbb{C}$ as before. The embedding $\pi$ induces $\langle\cdot,\cdot\rangle_\mathbf{h}$ on $\mathbf{h}_\mathbb{C}$ by:

$$\iota\,\langle a,b\rangle_\mathbf{h} = \langle\pi(a),\pi(b)\rangle_\mathbf{g} \,, \tag{4.1}$$

---

[3]Our metric is $\eta^{+-} = 2, \eta^{++} = \eta^{--} = 0$ and $d^2x = dx^+ \wedge dx^-$.

where $\iota$ is called the index of embedding [43]. In the following we take $\mathbf{g}_{\mathbb{C}}$ to be in the adjoint representation $R_{ad}$, this induces a representation $R_{ad} \circ \pi$ of $\mathbf{h}_{\mathbb{C}}$. Finally, we restrict ourselves to subgroups $H_{\mathbb{C}}$ for which the coset $G/H$ is a reductive homogenous space, that is we can take $\mathbf{h}_{\mathbb{C}}$ and $\mathbf{f}_{\mathbb{C}}$ in the decomposition $\mathbf{g}_{\mathbb{C}} = \mathbf{f}_{\mathbb{C}} \oplus \mathbf{h}_{\mathbb{C}}$ to satisfy

$$[\mathbf{h}_{\mathbb{C}}, \mathbf{h}_{\mathbb{C}}] \subset \mathbf{h}_{\mathbb{C}}, \quad [\mathbf{h}_{\mathbb{C}}, \mathbf{f}_{\mathbb{C}}] \subset \mathbf{f}_{\mathbb{C}}, \quad \langle \mathbf{h}_{\mathbb{C}}, \mathbf{f}_{\mathbb{C}} \rangle_{\mathbf{g}} = 0. \tag{4.2}$$

Since $\mathbf{f}_{\mathbb{C}}$ is orthogonal to $\mathbf{h}_{\mathbb{C}}$ by the final expression in (4.2) we call it the orthogonal complement.

Thus, given the two fields $A \in \mathbf{g}_{\mathbb{C}}$ and $B \in \mathbf{h}_{\mathbb{C}}$, we define the doubled four-dimensional Chern-Simons theory using the difference of two four-dimensional Chern-Simons actions, one for each field, and a new boundary term which couples $A$ and $B$ together:

$$S_{\mathrm{Dbld}}(A,B) = S_{\mathrm{4dCS}}(A) - S_{\mathrm{4dCS}}(B) + S_{\mathrm{bdry}}(A,B)$$

$$= \frac{1}{2\pi\hbar_{\mathbf{g}}} \int_{\Sigma \times C} \omega \wedge \left\langle A, dA + \frac{2}{3} A \wedge A \right\rangle_{\mathbf{g}} - \frac{1}{2\pi\hbar_{\mathbf{h}}} \int_{\Sigma \times C} \omega \wedge \left\langle B, dB + \frac{2}{3} B \wedge B \right\rangle_{\mathbf{h}}$$

$$- \frac{1}{2\pi\hbar_{\mathbf{h}}} \int_{\Sigma \times C} \omega \wedge \bar{\partial} \langle A|_{\mathbf{h}}, B \rangle_{\mathbf{h}}, \tag{4.3}$$

we will often refer to this action as the doubled theory for short. The final term in this action is the boundary term mentioned above, its only non-zero contributions are at the poles of $\omega$ and thus only modifies the defect equations of motion. The presence of $|_{\mathbf{h}}$ denotes the projection of $A$ onto $\mathbf{h}_{\mathbb{C}}$. so that $A|_{\mathbf{h}} \in \mathbf{h}_{\mathbb{C}}$. Likewise, $|_{\mathbf{f}}$ denotes the projection onto $\mathbf{f}_{\mathbb{C}}$. Later in this section we show the doubled action is gauge invariant if the two levels $\hbar_{\mathbf{g}}$ and $\hbar_{\mathbf{h}}$ satisfy:

$$\hbar_{\mathbf{g}} = \iota \hbar_{\mathbf{h}}. \tag{4.4}$$

For now, we use equations (4.1) and (4.4) to ensure our action contains a bilinear form and level and simplify our notation to: $\langle \cdot, \cdot \rangle_g = \langle \cdot, \cdot \rangle$ and $\hbar_{\mathbf{g}} = \hbar$. Upon doing this, we treat $B$ as a gauge field valued in $\mathbf{g}_{\mathbb{C}}$, whose components in $\mathbf{f}_{\mathbb{C}}$ vanish and drop the projection of $A$ in the boundary term since $\langle A|_{\mathbf{f}}, B \rangle$ vanishes by (4.2). Later, when discussing gauge invariance, we reintroduce the two bilinear forms and levels, and show (4.4) is necessary for the action to be gauge invariant.

Once a set of coordinates $z$ and $\bar{z}$ are chosen for $C$ it is clear the fields $A_z$ and $B_z$ fall out of the doubled action due to the wedge product with $\omega = \varphi(z)dz$ (since $dz \wedge dz = 0$). Thus the action is invariant under the additional gauge transformations:

$$A_z \longrightarrow A_z + \chi_z, \qquad B_z \longrightarrow B_z + \xi_z, \tag{4.5}$$

where $\chi_z$ and $\xi_z$ are arbitrary functions valued respectively in $\mathbf{g}_{\mathbb{C}}$ and $\mathbf{h}_{\mathbb{C}}$. Since $\chi_z$ and $\xi_z$ are arbitrary functions it follows that all field configurations of $A_z$ and $B_z$ are gauge equivalent, thus we work in the gauge $A_z = B_z = 0$ while $A$ and $B$ are:

$$A = A_+ dx^+ + A_- dx^- + A_{\bar{z}} d\bar{z}, \qquad B = B_+ dx^+ + B_- dx^- + B_{\bar{z}} d\bar{z}. \tag{4.6}$$

As in four-dimensional Chern-Simons the doubled action (4.3) is topological in $\Sigma$, but is not in $C$ since $\varphi(z)$ does not transform as a vector. The diffeomorphisms of $C$ which leave (4.3) invariant are those which leave $\omega$ invariant: if $z \to w(z)$ is a diffeomorphism then $\omega$ is invariant if $\varphi(w(z))\partial w/\partial z = \varphi(z)$.

The equations of motion of the doubled action (4.3) are found from the variation

$$\delta S_{\mathrm{Dbld}}(A,B) = \frac{1}{2\pi\hbar} \int_{\Sigma \times C} \omega \wedge \langle 2F(A), \delta A \rangle - \frac{1}{2\pi\hbar} \int_{\Sigma \times C} \omega \wedge \langle 2F(B), \delta B \rangle \tag{4.7}$$

$$- \frac{1}{2\pi\hbar} \int_{\Sigma \times C} \bar{\partial}\omega \wedge \langle A - B, \delta A + \delta B \rangle.$$

The bulk equations of motion are:

$$\omega \wedge F(A) = 0 \,, \qquad \omega \wedge F(B) = 0 \,, \tag{4.8}$$

while the defect equation of motion is:

$$\frac{1}{2\pi\hbar} \int_{\Sigma \times C} \bar{\partial}\omega \wedge \langle A - B, \delta A + \delta B \rangle = 0 \,. \tag{4.9}$$

In the next section we discuss the solutions to this equation.

As a final remark, we note the following: when deriving (4.9) we sent a total derivative to zero implicitly assuming $\omega \wedge (A - B) \wedge (\delta A + \delta B)$ contain no poles. If $\zeta$ is a zero of $\omega$ of order $m_\zeta$ and $k_\zeta^\pm$, $h_\zeta^\pm$ and $\kappa_\zeta^\pm$ denote the orders of poles of $A_\pm|_{\mathbf{f}}$, $A_\pm|_{\mathbf{h}}$ and $B_\pm$ respectively, then this requires the following inequalities be satisfied:

$$k_\zeta^+ + k_\zeta^- \leqslant m_\zeta \,, \qquad h_\zeta^+ + h_\zeta^- \leqslant m_\zeta \,, \qquad h_\zeta^+ + \kappa_\zeta^- \leqslant m_\zeta \,, \tag{4.10}$$
$$\kappa_\zeta^+ + h_\zeta^- \leqslant m_\zeta \,, \qquad \kappa_\zeta^+ + \kappa_\zeta^- \leqslant m_\zeta \,.$$

## 4.1 Boundary Conditions and Gauged Type B Defects

In standard four-dimensional Chern-Simons, the defect equations of motion require boundary conditions on $A$ at the poles of $\omega$, which insert type B defects. Similarly, in the doubled theory the defect equations of motion (4.9) require boundary conditions on $A$ and $B$ at the poles of $\omega$, which introduce analogues of the type B defects which we call 'gauged' type B defects. On these defects we find the $H_\mathbb{C}$ symmetry of $B$ is gauged out of the $G_\mathbb{C}$ symmetry of $A$, introducing an $H_\mathbb{C}$ gauge symmetry in our sigma models. In the following we define the gauged type B defects for simple and double poles of $\omega$.

To solve (4.9) we use the decomposition $\mathbf{g}_\mathbb{C} = \mathbf{f}_\mathbb{C} \oplus \mathbf{h}_\mathbb{C}$ and the orthogonality of $\mathbf{f}_\mathbb{C}$ with respect to $\mathbf{h}_\mathbb{C}$ to separate (4.9) into a set of equations in $\mathbf{f}_\mathbb{C}$ and a set in $\mathbf{h}_\mathbb{C}$. After using (2.12) these equations are:

$$\sum_{q \in P} \sum_{l=0}^{n_q-1} \frac{\eta_q^l}{l!} \epsilon^{jk} \partial_z^l \langle A_j|_{\mathbf{f}}, \delta A_k|_{\mathbf{f}} \rangle |_{(q,\bar{q})} = 0 \,, \tag{4.11}$$

$$\sum_{q \in P} \sum_{l=0}^{n_q-1} \frac{\eta_q^l}{l!} \epsilon^{jk} \partial_z^l \langle A_j|_{\mathbf{h}} - B_j, \delta A_k|_{\mathbf{h}} + \delta B_k \rangle |_{(q,\bar{q})} = 0 \,, \tag{4.12}$$

where $i, j = \pm$, $P$ the set of all poles of $\omega$ and $\eta_q^l$ the residue defined in (2.4). Note, we have dropped the integral over $\Sigma$ as the boundary conditions we construct ensure the integrand, and thus integral, vanish. We solve these equations individually at each pole of $\omega$, hence our defect equations of motion for an $\omega$ with at most double poles reduce to:

$$\left(\eta_q^0 + \eta_q^1 \partial_z\right) \epsilon^{jk} \langle A_j|_{\mathbf{f}}, \delta A_k|_{\mathbf{f}} \rangle |_{(q,\bar{q})} = 0 \,, \tag{4.13}$$
$$\left(\eta_q^0 + \eta_q^1 \partial_z\right) \epsilon^{jk} \langle A_j|_{\mathbf{h}} - B_j, \delta A_k|_{\mathbf{h}} + \delta B_k \rangle |_{(q,\bar{q})} = 0 \,, \tag{4.14}$$

where $\eta_q^1 = 0$ for simple poles. Before discussing solutions to these equations we emphasise that boundary conditions hold in all gauges, thus in the gauge transformations:

$$A \longrightarrow A^u = u(d + A)u^{-1} \,, \qquad B \longrightarrow B^v = v(d + B)v^{-1} \,, \tag{4.15}$$

the group elements $u \in G_\mathbb{C}$ and $v \in H_\mathbb{C}$ are constrained to ensure this is the case. It is necessary to include these constraints in our boundary conditions as they will be used to prove the action is gauge invariant under large gauge transformations in the next subsection.

## Gauged Chiral Boundary Conditions

The Gauged chiral boundary condition is a solution to (4.13) and (4.14) for simple poles of $\omega$, thus our defect equations of motion are:

$$\epsilon^{jk}\left\langle A_j|_{\mathbf{f}},\delta A_k|_{\mathbf{f}}\right\rangle|_{(q,\bar{q})}=0\,,\qquad \epsilon^{jk}\left\langle A_j|_{\mathbf{h}}-B_j,\delta A_k|_{\mathbf{h}}+\delta B_k\right\rangle|_{(q,\bar{q})}=0\,,\tag{4.16}$$

where we have dropped an $\eta_q^0$ as it is an arbitrary overall constant. The gauged chiral boundary condition at a pole $q$ is the solution:

$$A_-|_{\mathbf{f}}=0\,,\qquad A_\pm|_{\mathbf{h}}-B_\pm=0\,,\tag{4.17}$$

where the first condition implies $\delta A_-|_{\mathbf{f}}=0$ and thus solves the first of equations (4.16).

These boundary conditions must be preserved by the gauge transformation (4.15), which occurs if the following conditions hold at the pole $q$:

$$(uB_-u^{-1}+u\partial_-u^{-1})|_{\mathbf{f}}=0\,,\tag{4.18}$$

$$(y^{-1}A_\pm y+y^{-1}\partial_\pm y-A_\pm)|_{\mathbf{h}}=0\,,\quad\text{where}\quad y=u^{-1}v\,.\tag{4.19}$$

This first condition follows from a gauge transformation of (4.17) which under (4.15) transforms as:

$$A_-|_{\mathbf{f}}\longrightarrow A_-^u|_{\mathbf{f}}=(uB_-u^{-1}+u\partial_-u^{-1})|_{\mathbf{f}}=0\,,\tag{4.20}$$

where in the first equality we have used (4.17), while the second equality is the statement that we preserve the boundary condition. Similarly, the second equation in (4.17) transforms as:

$$A_\pm|_{\mathbf{h}}-B_\pm\longrightarrow A_\pm^u|_{\mathbf{h}}-B_\pm^v=(uA_\pm u^{-1}+u\partial_\pm u^{-1})|_{\mathbf{h}}-vB_\pm v^{-1}-v\partial_\pm v^{-1}=0\,.\tag{4.21}$$

Let $y=u^{-1}v\in G_\mathbb{C}$ be the group element which $u$ and $v$ differ by at $q$ and substitute in $u=vy^{-1}$. Thus this equation becomes:

$$v(y^{-1}A_\pm y+y^{-1}\partial_\pm y-B_\pm)v^{-1}|_{\mathbf{h}}=0\,.\tag{4.22}$$

Since $v\in H$ and $[\mathbf{h}_\mathbb{C},\mathbf{f}_\mathbb{C}]\subset\mathbf{f}_\mathbb{C}$ then $v\mathbf{f}_\mathbb{C}v^{-1}=\mathbf{f}_\mathbb{C}$ and similarly $v\mathbf{h}_\mathbb{C}v^{-1}=\mathbf{h}_\mathbb{C}$. Hence (4.22) is equivalent to

$$(y^{-1}A_\pm y+y^{-1}\partial_\pm y-A_\pm)|_{\mathbf{h}}=0\,,\tag{4.23}$$

where we have used $A_\pm|_{\mathbf{h}}=B_\pm$.

Altogether, the gauged chiral boundary condition consists of the boundary conditions

$$A_-|_{\mathbf{f}}=0\,,\qquad A_\pm|_{\mathbf{h}}-B_\pm=0\,,\tag{4.24}$$

and the constraints

$$(uB_-u^{-1}+u\partial_-u^{-1})|_{\mathbf{f}}=0\,,\quad(y^{-1}A_\pm y+y^{-1}\partial_\pm y-A_\pm)|_{\mathbf{h}}=0\,.\tag{4.25}$$

## Gauged Anti-Chiral Boundary Conditions

We also define the gauged anti-chiral boundary condition at a simple pole of $\omega$ as the boundary conditions

$$A_+|_{\mathbf{f}}=0\,,\qquad A_\pm|_{\mathbf{h}}-B_\pm=0\,,\tag{4.26}$$

and the constraints

$$(uB_+u^{-1}+u\partial_+u^{-1})|_{\mathbf{f}}=0\,,\tag{4.27}$$

$$(y^{-1}A_\pm y+y^{-1}\partial_\pm y-A_\pm)|_{\mathbf{h}}=0\,,\tag{4.28}$$

where $y=u^{-1}v$.

**Gauged Dirichlet Boundary Conditions**

At double poles of $\omega$ the defect equations of motion (4.13) and (4.14) are:

$$\left(\eta_q^0 + \eta_q^1 \partial_z\right) \epsilon^{jk} \left\langle A_j|_{\mathbf{f}}, \delta A_k|_{\mathbf{f}} \right\rangle |_{(q,\bar{q})} = 0\,, \tag{4.29}$$

$$\left(\eta_q^0 + \eta_q^1 \partial_z\right) \epsilon^{jk} \left\langle A_j|_{\mathbf{h}} - B_j, \delta A_k|_{\mathbf{h}} + \delta B_k \right\rangle |_{(q,\bar{q})} = 0\,. \tag{4.30}$$

One class of solutions to this equation are the gauged Dirichlet boundary conditions:

$$A_\pm|_{\mathbf{f}} = 0\,, \qquad A_\pm|_{\mathbf{h}} - B_\pm = 0\,, \qquad \partial_z(A_\pm|_{\mathbf{h}} - B_\pm) = 0\,, \tag{4.31}$$

$$A_{\bar{z}}|_{\mathbf{h}} - B_{\bar{z}} = 0 \qquad \partial_z(A_{\bar{z}}|_{\mathbf{h}} - B_{\bar{z}}) = 0\,.$$

One might wonder where these final two conditions come from. The first is to ensure $\psi$ in (2.11) is regular, thus ensuring a regular doubled action in the presence of double poles. The second is ensure that when the action is reduced to the type B defects the resultant theory is genuinely two dimensional, depending only upon $x^\pm$. We will expand on this point in section 5.4. By arguments similar to those used for the gauged chiral boundary condition the first two conditions are preserved by gauge transformations which satisfy:

$$(uB_\pm u^{-1} + u\partial_\pm u^{-1})|_{\mathbf{f}} = 0\,, \tag{4.32}$$

$$(y^{-1}A_\pm y + y^{-1}\partial_\pm y - A_\pm)|_{\mathbf{h}} = 0\,, \quad \text{where} \quad y = u^{-1}v\,. \tag{4.33}$$

while the third condition is preserved if:

$$\partial_z(y^{-1}A_\pm y + y^{-1}\partial_\pm y - A_\pm)|_{\mathbf{h}} = 0\,, \tag{4.34}$$

which has been derived using (4.33).

## 4.2   Gauge Invariance

In this section we prove that the doubled action (4.3) is gauge invariant for field configurations which satisfy any of the boundary conditions defined in the previous section at the poles of $\omega$. We reintroduce the trace $\langle \cdot, \cdot \rangle_{\mathbf{g}} = \langle \cdot, \cdot \rangle$ and $\langle \cdot, \cdot \rangle_{\mathbf{h}}$ into the action and show the action is gauge invariant if $\hbar_{\mathbf{g}} = \iota\hbar_{\mathbf{h}}$ after having used $\mathrm{Tr}_{\mathbf{h}} = \iota\mathrm{Tr}_{\mathbf{g}}$, (4.1). As a reminder, the gauge transformations of the gauge fields $A$ and $B$ are:

$$A \longrightarrow A^u = u(d + A)u^{-1}\,, \qquad B \longrightarrow B^v = v(d + B)v^{-1}\,, \tag{4.35}$$

where $y = u^{-1}v \in G_{\mathbb{C}}$.

Under the gauge transformations (4.35), the action transforms as:

$$S_{\mathrm{Dbld}}(A, B) \longrightarrow S_{\mathrm{Dbld}}(A^u, B^v) = S_{\mathrm{4dCS}}(A^u) - S_{\mathrm{4dCS}}(B^v) + S_{\mathrm{Bdry}}(A^u, B^v)\,, \tag{4.36}$$

where:

$$S_{\mathrm{4dCS}}(A^u) = S_{\mathrm{4dCS}}(A) + \frac{1}{2\pi\hbar_{\mathbf{g}}} \int_{\Sigma \times C} \omega \wedge \bar{\partial} \left\langle u^{-1}du, A \right\rangle_{\mathbf{g}} + \frac{1}{6\pi\hbar_{\mathbf{g}}} \int_{\Sigma \times C} \omega \wedge \left\langle u^{-1}du, (u^{-1}du)^2 \right\rangle_{\mathbf{g}}\,, \tag{4.37}$$

$$S_{\mathrm{Bdry}}(A^u, B^v) = -\frac{1}{2\pi\hbar_{\mathbf{h}}} \int_{\Sigma \times C} \omega \wedge \bar{\partial} \left\langle uAu^{-1} - duu^{-1}, vBv^{-1} - dvv^{-1} \right\rangle_{\mathbf{h}}\,. \tag{4.38}$$

In these expressions we have sent a total derivative in $d_\Sigma$ to zero by requiring $A$, $B$, $u$ and $v$ die off sufficiently fast enough at infinity. Upon using the Polyakov-Wiegmann identity [44]:

$$\left\langle u^{-1}du, (u^{-1}du)^2 \right\rangle - \left\langle v^{-1}dv, (v^{-1}dv)^2 \right\rangle = \left\langle ydy^{-1}, (ydy^{-1})^2 \right\rangle + 3d \left\langle vdv^{-1}, duu^{-1} \right\rangle\,, \tag{4.39}$$

along with (4.1) and (4.4), the two Wess-Zumino terms in (4.36) can be written as:

$$\frac{1}{6\pi\hbar_{\mathbf{g}}} \int_{\Sigma\times C} \omega \wedge \left\langle u^{-1}du, (u^{-1}du)^2 \right\rangle_{\mathbf{g}} - \frac{1}{6\pi\hbar_{\mathbf{h}}} \int_{\Sigma\times C} \omega \wedge \left\langle v^{-1}dv, (v^{-1}dv)^2 \right\rangle_{\mathbf{h}} \tag{4.40}$$
$$= \frac{1}{6\pi\hbar} \int_{\Sigma\times C} \omega \wedge \left\langle ydy^{-1}, (ydy^{-1})^2 \right\rangle + \frac{1}{2\pi\hbar} \int_{\Sigma\times C} \omega \wedge \bar{\partial} \left\langle vdv^{-1}, duu^{-1} \right\rangle ,$$

where we have used (4.1) and then set $\hbar_{\mathbf{g}} = \hbar$.

Using $u = vy^{-1}$ and cancelling several terms, this reduces to:

$$S_{\text{Dbld}}(A^u, B^v) = S_{\text{4dCS}}(A) - S_{\text{4dCS}}(B) + \frac{1}{6\pi\hbar} \int_{\Sigma\times C} \omega \wedge \left\langle ydy^{-1}, ydy^{-1} \wedge ydy^{-1} \right\rangle$$
$$+ \frac{1}{2\pi\hbar} \int_{\Sigma\times C} \omega \wedge \bar{\partial} \left[ \left\langle ydy^{-1}, A \right\rangle - \left\langle y^{-1}Ay + y^{-1}dy, B \right\rangle \right] . \tag{4.41}$$

In the previous subsection we proved gauged boundary conditions imply the following equation holds at the poles of $\omega$:

$$(y^{-1}A_{\pm}y + y^{-1}\partial_{\pm}y)|_{\mathbf{h}} = A_{\pm}|_{\mathbf{h}}, \qquad \partial_z(y^{-1}A_{\pm}y + y^{-1}\partial_{\pm}y)|_{\mathbf{h}} = \partial_z A_{\pm}|_{\mathbf{h}} \tag{4.42}$$

with the second applying to double poles only. Using these, we can reduce the final term of (4.41) to $S_{\text{Bdry}}(A, B)$ and thus find:

$$S_{\text{Dbld}}(A^u, B^v) = S_{\text{Dbld}}(A, B) + I_2 + I_3 , \tag{4.43}$$

$$I_2 = \frac{1}{2\pi\hbar} \int_{\Sigma\times C} \bar{\partial}\omega \wedge \left\langle ydy^{-1}, A \right\rangle , \tag{4.44}$$

$$I_3 = \frac{1}{6\pi\hbar} \int_{\Sigma\times C} \omega \wedge \left\langle ydy^{-1}, ydy^{-1} \wedge ydy^{-1} \right\rangle . \tag{4.45}$$

In the following we focus on sigma models which are recovered from doubled four-dimensional Chern-Simons on $\mathbb{R}^2 \times \mathbb{C}P^1$, thus as described in appendix B and section 2.4 the three-form $\left\langle ydy^{-1}, ydy^{-1} \wedge ydy^{-1} \right\rangle$ is exact. This allows us to write the Wess-Zumino term of (4.45) as:

$$I_3 = \frac{1}{6\pi\hbar} \int_{\Sigma\times C} \bar{\partial}\omega \wedge E(y) . \tag{4.46}$$

Since $d(ydy^{-1})^2 = 0$ identically it is clear that $E(y) \neq \left\langle ydy^{-1}, ydy^{-1} \right\rangle$ and thus that $I_2$ and $I_3$ must vanish independently, as we now show. As in section 2.4 we achieve this by asking that the terms in $I_2$ and $I_3$ both vanish at each pole $q$ separately.

### 4.2.1   $I_3$, The Wess-Zumino Term:

As in section 2.4 the $\xi_{+-}dx^+ \wedge dx^-$ is the component of $E(y)$ which contributes to the Wess-Zumino term. Using the coordinates on the group $G_{\mathbb{C}}$, $\zeta^a$, we have the expressions $\zeta_{+-} = f(\zeta)_{ab}\partial_+\zeta^a\partial_-\zeta^b$ and $\partial_{\pm}y = \partial_{\pm}\zeta^a\partial_a y$.

For simple poles $E(y)$ need only be at least linear in $z$ for the contribution to vanish.

- For gauged chiral boundary conditions, the field $A_+$ generates a chiral Kac-Moody current on the defect whose symmetry we wish to preserve. Therefore, we do not impose any further constraints upon $\partial_+y$. Thus, to ensure gauge invariance we demand:

$$\partial_-y = 0 , \qquad y^{-1}B_-y|_{\mathbf{h}} - B_- = 0 , \tag{4.47}$$

where the second equation follows from (4.42) and the imposition of $A_- = B_-$. By $\partial_- y = \partial_- \zeta^a \partial_a y$ it follows that $\partial_- \zeta = 0$ ensuring the Wess-Zumino term vanishes.

- For gauged anti-chiral boundary conditions, similarly, since $A_-$ produces an anti-chiral current on the defect, we wish to avoid impose any further conditions upon $\partial_- y$, thus we demand:

$$\partial_+ y = 0, \qquad y^{-1} B_+ y|_\mathbf{h} - B_+ = 0. \tag{4.48}$$

This ensures gauge invariance by an analogous argument to that used for gauged chiral boundary conditions.

- For gauged Dirichlet boundary conditions, the contribution from double poles vanishes if $E(y) = 0$ and $\partial_z E(y) = 0$, thus we demand:

$$\partial_\pm y = 0, \quad \partial_z (\partial_\pm y) = 0,$$
$$y^{-1} A_\pm y|_\mathbf{h} = B_\pm, \quad \partial_z (y^{-1} A_\pm y|_\mathbf{h}) = \partial_z B_\pm. \tag{4.49}$$

### 4.2.2 $I_2$

The contribution to $I_2$ from the pole at $q$ is

$$I_2^q = \frac{1}{2\pi\hbar} \sum_{l=0}^{n_q - 1} \int_{\Sigma_q} d^2 x \, \frac{\eta_q^l}{l!} \epsilon^{jk} \partial_z^l \left\langle y\partial_j y^{-1}, A_k \right\rangle. \tag{4.50}$$

- Gauged Chiral/Anti-Chiral Boundary Conditions:

  At a simple pole, (4.50) is:

$$I_{2,\text{sing}}^q = \frac{1}{2\pi\hbar} \eta_q^0 \int_{\Sigma_q} d^2 x \, \epsilon^{jk} \left\langle y\partial_j y^{-1}, A_k \right\rangle. \tag{4.51}$$

  Upon imposing the gauged chiral boundary conditions (4.17) and (4.47) this expression reduces to:

$$I_{2,\text{sing}}^q = \frac{1}{2\pi\hbar} \eta_q^0 \int_{\Sigma_q} d^2 x \left\langle y\partial_+ y^{-1}, B_- \right\rangle, \tag{4.52}$$

  where we have used $\epsilon^{+-} = 1$. If we require that $y\partial_+ y^{-1} \in \mathbf{f}_\mathbb{C}$ at $z = q$ then this term vanishes by $\langle \mathbf{f}_\mathbb{C}, \mathbf{h}_\mathbb{C} \rangle = 0$, thus $I_{\text{sing}} = 0$ for gauged chiral boundary conditions. The same argument follows for gauged anti-chiral boundary conditions after one swaps $+$ for $-$ in the above and uses the gauged anti-chiral boundary conditions.

- Gauged Dirichlet Boundary Conditions

  At a double pole, $I_2^q$ is:

$$I_{2,\text{dble}}^q = \frac{1}{2\pi\hbar} \left( \eta_q^0 + \eta_q^1 \partial_z \right) \int_{\Sigma_q} d^2 x \epsilon^{jk} \left\langle y\partial_j y^{-1}, A_k \right\rangle. \tag{4.53}$$

  By imposing (4.49), we find that $\epsilon^{jk} \left\langle y\partial_j y^{-1}, A_k \right\rangle$ and its $z$ derivative are zero implying that $I_{\text{dble}}^q$ vanishes.

It follows from the above analysis that the doubled Chern-Simons action is indeed gauge invariant.

# 5 The Unified Gauged Sigma Model

In this section we reduce the doubled action (4.3) to a unified gauged sigma model following arguments similar to those used in [25]. As in [25] one constructs field configurations $A$ and $B$ of the doubled equations of motion (4.8) which are gauge equivalent to two Lax connection $\mathcal{L}_A$ and $\mathcal{L}_B$. Using the unified gauged model a set of field configurations (and thus Lax connections) determines a gauged sigma model whose integrability is determined by the existence of the Lax connections.

We begin by introducing two classes of group elements, $\{\hat{g}\}$ and $\{\hat{h}\}$, using them to rewrite the doubled action. We prove that one can construct group elements which satisfy the archipelago conditions of [25] which are used to reduce the rewritten doubled action to a two-dimensional theory which sits on the defects at the poles of $\omega$. By varying this action we show $A$ and $B$ are gauge equivalent to Lax connections. Finally, we construct several examples of gauged sigma models from the unified gauged sigma model. In this section we fix $C = \mathbb{C}P^1$ with the coordinates $z$ and $\bar{z}$.

## 5.1 Gauge fields and group elements

As in section 3.1, the fields $\overline{A} = A_{\bar{z}} d\bar{z}$ and $\overline{B} = B_{\bar{z}} d\bar{z}$ can be expressed in terms of group elements $\hat{g} : \Sigma \times \mathbb{C}P^1 \to G_{\mathbb{C}}$ and $\hat{h} : \Sigma \times \mathbb{C}P^1 \to H_{\mathbb{C}}$ by:

$$\overline{A} = \hat{g}\bar{\partial}\hat{g}^{-1} = -(\bar{\partial}\hat{g})\hat{g}^{-1}, \qquad \overline{B} = \hat{h}\bar{\partial}\hat{h}^{-1} = -(\bar{\partial}\hat{h})\hat{h}^{-1}. \tag{5.1}$$

These define equivalence classes of elements, $\{\hat{g}\}$ and $\{\hat{h}\}$, related by right-multiplication by group elements that are independent of $\bar{z}$ (which we call right-redundancy). We again will only consider forms $\omega$ with a pole at infinity and choose canonical representatives of $\hat{g}$ and $\hat{h}$ which are the identity at $z = \infty$, and from now on the notation $\hat{g}$ and $\hat{h}$ will always imply this canonical representative. As in the section 3.3, gauge transformations on $A$ and $B$ by $u$ and $v$ manifest as the following transformations on $\hat{g}$ and $\hat{h}$:

$$A \longrightarrow A' = uAu^{-1} + udu^{-1} \qquad\qquad B \longrightarrow B' = vBv^{-1} + vdv^{-1} \tag{5.2}$$

$$\hat{g} \longrightarrow \hat{g}' = u\hat{g}u_\infty^{-1}, \qquad\qquad \hat{h} \longrightarrow \hat{h}' = v\hat{h}v_\infty^{-1}, \tag{5.3}$$

where $u_\infty$ and $v_\infty$ are the values of $y$ and $v$ at $z = \infty$, and these terms are included to ensure the group elements satisfy $\hat{g}'|_{(\infty,\infty)} = \hat{h}'|_{(\infty,\infty)} = 1$.

## 5.2 More Lax Connections

A connection $\mathcal{L}$ is a Lax connection if it satisfies properties $1 - 3$ in section 3.3. Using the group elements $\hat{g}$ and $\hat{h}$ one can construct the fields $\mathcal{L}_A$ and $\mathcal{L}_B$ from $A$ and $B$ which satisfy the conditions required of a Lax connection. By gauge transforming $A$ by $\hat{g}^{-1}$ and $B$ by $\hat{h}^{-1}$ one finds:

$$\mathcal{L}_A = \hat{g}^{-1}d\hat{g} + \hat{g}^{-1}A\hat{g}, \qquad \mathcal{L}_B = \hat{h}^{-1}d\hat{h} + \hat{h}^{-1}B\hat{h}, \tag{5.4}$$

where $\overline{\mathcal{L}}_A = \overline{\mathcal{L}}_B = 0$ follows from equation (5.1).

From equations (5.2) and (5.3), gauge transformations on $A$ and $B$ by $u$ and $v$ are equivalent to the following changes to $\mathcal{L}_A$ and $\mathcal{L}_B$:

$$\mathcal{L}_A \longrightarrow \mathcal{L}_{A'} = u_\infty \mathcal{L}_A u_\infty^{-1} + u_\infty du_\infty^{-1}, \tag{5.5}$$

$$\mathcal{L}_B \longrightarrow \mathcal{L}_{B'} = v_\infty \mathcal{L}_B v_\infty^{-1} + v_\infty dv_\infty^{-1}, \tag{5.6}$$

where $u_\infty = u|_{(\infty,\infty)}$ and $v_\infty = v|_{(\infty,\infty)}$.

The equations of motion for $A$ and $B$ (4.8) imply:

$$\omega \wedge \bar{\partial} \mathcal{L}_A = 0, \qquad d_\Sigma \mathcal{L}_A + \mathcal{L}_A \wedge \mathcal{L}_A = 0, \tag{5.7}$$

$$\omega \wedge \bar{\partial} \mathcal{L}_B = 0, \qquad d_\Sigma \mathcal{L}_B + \mathcal{L}_B \wedge \mathcal{L}_B = 0. \tag{5.8}$$

The first and third equations mean $\mathcal{L}_A$ and $\mathcal{L}_B$ have a meromorphic dependence upon $z$ and thus satisfy the second property of a Lax connection. The second and fourth equations mean $\mathcal{L}_A$ and $\mathcal{L}_B$ are flat in the plane $\Sigma$ and we will demonstrate in the following that these are the equations of motion of our sigma model, ensuring we satisfy the first property of a Lax connection.

Following exactly the same arguments as in section 3.3 for $I = A, B$, in the case where $\Sigma = S^1 \times \mathbb{R}$ we construct monodromy matrices $U_i$:

$$U_A(z,t) = P \exp\left( \int_0^{2\pi} \mathcal{L}_{A\,\theta} d\theta \right) = \hat{g}^{-1} P \exp\left( \int_0^{2\pi} A_\theta d\theta \right) \hat{g}, \tag{5.9}$$

$$U_B(z,t) = P \exp\left( \int_0^{2\pi} \mathcal{L}_{B\,\theta} d\theta \right) = \hat{h}^{-1} P \exp\left( \int_0^{2\pi} B_\theta d\theta \right) \hat{h} \tag{5.10}$$

to find conserved quantities $W_I$ whose coefficients are conserved charges.

## 5.3 The Archipelago Conditions

In this section we extend the definition of the archipelago gauge for the Four-Dimensional Chern-Simons Theory to the doubled model. This is very straightforward, as it is simply that the group elements $\hat{g}$ and $\hat{h}$ defined in section 5.1 must satisfy

(a) $\hat{g}$ satisfies the archipelago conditions of section 3.2

(b) $\hat{h}$ is the identity.

These conditions are equivalent to a gauge choice on $A$ and $B$. They allow one to reduce the doubled action to a unified gauged sigma model on the defects at the poles of $\omega$ - we leave this to the next section.

We show that the archipelago conditions are a gauge choice by exhibiting a gauge transformation which will put any gauge field configuration into this gauge. If $A$ and $B$ define group elements $\hat{g}$ and $\hat{h}$, then this is equivalent to finding gauge transformations $U \in G$ and $V \in H$ consistent with the defect boundary conditions such that $\tilde{g} = U\hat{g}U_\infty^{-1}$ satisfies (a) and $\tilde{h} = Vh V_\infty^{-1} = 1$ satisfies (b).

To show that the gauge transformation of $(A, B)$ by $(U, V)$ to the archipelago gauge preserves the defect boundary conditions, we shall split it into two consecutive gauge transformations[4]

$$(\hat{g}, \hat{h}) \xrightarrow{(\hat{h}^{-1}, \hat{h}^{-1})} (\hat{h}^{-1}\hat{g}, 1) \equiv (\hat{g}^h, 1) \xrightarrow{(\hat{u}, 1)} (\hat{u}g^h, 1) \equiv (\tilde{g}, 1), \tag{5.11}$$

where we have used the facts that $\hat{h}_\infty = \hat{u}_\infty = 1$.

In the first step we use the fact that gauge transformations of $B$ are unrestricted by the three boundary conditions we have defined above, so we can simply choose $v = \hat{h}^{-1}$. This takes $B$ everywhere in $\mathbb{C}P^1$ to:

$$B_{\bar{z}} = 0, \qquad B_\Sigma = \mathcal{L}_B. \tag{5.12}$$

For this gauge transformation to be consistent with gauged chiral boundary conditions it must satisfy (4.18) and (4.19). By noting that $\hat{h} \in H_\mathbb{C}$ it follows that $h^{-1}\partial_-\hat{h}|_{\mathbf{f}} = 0$ and thus that (4.18) reduces to:

$$\hat{h}^{-1}A_-\hat{h}|_{\mathbf{f}} = 0, \tag{5.13}$$

---

[4]Note that $\hat{g}^h \equiv \hat{h}^{-1}\hat{g}$ and does not denote conjugation by $h$, as is often meant by this notation

which follows from the boundary condition $A_- = B_-$ as $\hat{h}^{-1}B_-\hat{h} \in \mathbf{h}_\mathbb{C}$. This argument also holds for both gauged anti-chiral and Dirichlet boundary conditions by substituting $-$ for $+$ in the above argument. Since $y = u^{-1}v = 1$ it is also clear that:

$$(y^{-1}\partial_\pm y + y^{-1}A_\pm y - A_\pm)|_\mathbf{h} = 0, \qquad \partial_z(y^{-1}\partial_\pm y + y^{-1}A_\pm y - A_\pm)|_\mathbf{h} = 0, \tag{5.14}$$

thus satisfying the required conditions on $y$.

For the second step, we first define $\tilde{g}$ using exactly the same argument as in section 3.2, and then set $\tilde{u} = \tilde{g}(\hat{g}^h)^{-1}$. We now show that this third gauge transformation preserves each of the defect boundary conditions in turn.

### Consistency with Gauged Chiral/Anti-Chiral Boundary Conditions

This proof is essentially unchanged from the one given in section 3.2. As constructed $\tilde{g} = \hat{g}^h$ at the poles $q \in P$ implying $y^{-1} = \tilde{u} = \tilde{g}(\hat{g}^h)^{-1} = 1$, therefore the required boundary conditions:

$$(\tilde{u}B_\pm\tilde{u}^{-1} + \tilde{u}\partial_\pm\tilde{u}^{-1})|_\mathbf{f} = 0, \tag{5.15}$$

$$(y^{-1}A_\pm y - y^{-1}\partial_\pm y - A_\pm)|_\mathbf{h} = 0, \tag{5.16}$$

are obviously satisfied.

### Consistency with Gauged Dirichlet Boundary Conditions

The first two boundary conditions which $\tilde{u} = \tilde{g}(\hat{g}^h)^{-1}$ must satisfy are (5.15) and (5.16), these are satisfied for the same reason as the gauged chiral/anti-chiral boundary conditions. This leaves the condition:

$$\partial_z(\tilde{u}A_\pm\tilde{u}^{-1} + \tilde{u}\partial_\pm\tilde{u}^{-1} - A_\pm)|_\mathbf{h} = 0, \tag{5.17}$$

where we've used $y = \tilde{u}^{-1}$ since $v = 1$. The final two boundary conditions of (4.31) in the gauge $B_{\bar{z}} = 0$ reduce to $A_{\bar{z}}|_\mathbf{h} = 0$ and $\partial_z A_{\bar{z}}|_\mathbf{h} = 0$, which since $A_{\bar{z}} = \hat{g}^h\partial_{\bar{z}}(\hat{g}^h)^{-1}$ imply $\hat{g}^h\partial_z(\hat{g}^h)^{-1}|_\mathbf{h} = \hat{g}^h\partial_{\bar{z}}(\hat{g}^h)^{-1}|_\mathbf{h} = 0$. These conditions together with the third archipelago condition imply $\tilde{u}^{-1}\partial_z\tilde{u} = 0$, hence at the pole $q$ it follows $\partial_z(\tilde{u}\partial_\pm\tilde{u}^{-1})|_\mathbf{h} = 0$ and $\partial_z(\tilde{u}A_\pm\tilde{u}^{-1})|_\mathbf{h} = A_\pm|_\mathbf{h}$. Thus the condition is satisfied.

In [39] an alternative approach to the archipelago conditions was used to localise the theory to the defects. In their construction one has fewer boundary conditions on $A$ meaning a larger set of bundles are considered than in our construction. However, because these field configurations have fewer restrictions they also possess more gauge symmetry. As a result we expect the two constructions to be equivalent, realising the same Lax connections and integrable models.

The result is that, in the archipelago gauge, $\tilde{h} = 1$, $\bar{B} = 0$, and (combined with the results from section 5.4) the gauge fields take the form

$$A = \tilde{g}d\tilde{g}^{-1} + \tilde{g}\mathcal{L}_A\tilde{g}^{-1}, \qquad B = \mathcal{L}_B. \tag{5.18}$$

At a pole $q$ of $\omega$, these imply

$$\mathcal{L}_A|_{(q,\bar{q})} = g_q^{-1}A|_{(q,\bar{q})}g_q + g_q^{-1}dg_q, \qquad \mathcal{L}_B|_{(q,\bar{q})} = B|_{(q,\bar{q})} \tag{5.19}$$

## 5.4 The Unified Gauged Sigma Model Action

In this subsection we use the archipelago conditions to localise the doubled action (4.3) to a two-dimensional action the defects at the poles of $\omega$, as was done for the standard four-dimensional Chern-Simons action in [25]. We then show that the equations of motion imply that $\mathcal{L}_A$ and $\mathcal{L}_B$ are meromorphic and flat, and that

the gauged boundary conditions imply $(g_q d_\Sigma g_q^{-1} + g_q \mathcal{L}_A g_q^{-1})|_\mathbf{h} = \mathcal{L}_B$ We will conclude this subsection by construction the gauged WZW model as an example.

We start by assuming the gauge fields are in the archipelago gauge, so that they take the form (5.1). Substituting these expressions into the doubled action. we find:

$$S_{\text{Dbld}}(A, B) = \frac{1}{2\pi\hbar} \int_{\Sigma \times \mathbb{C}P^1} \omega \wedge \text{Tr} \left( \mathcal{L}_A \wedge \bar{\partial} \mathcal{L}_A \right) - \frac{1}{2\pi\hbar} \int_{\Sigma \times \mathbb{C}P^1} \bar{\partial}\omega \wedge \text{Tr} \left( \mathcal{L}_A \wedge \tilde{g}^{-1} d_\Sigma \tilde{g} \right) \tag{5.20}$$

$$- \frac{1}{2\pi\hbar} \int_{\Sigma \times \mathbb{C}P^1} \omega \wedge \text{Tr} \left( \mathcal{L}_B \wedge \bar{\partial} \mathcal{L}_B \right) + \frac{1}{6\pi\hbar} \int_{\Sigma \times \mathbb{C}P^1} \omega \wedge \text{Tr} \left( \tilde{g}^{-1} d\tilde{g} \wedge \tilde{g}^{-1} d\tilde{g} \wedge \tilde{g}^{-1} d\tilde{g} \right)$$

$$- \frac{1}{2\pi\hbar} \int_{\Sigma \times \mathbb{C}P^1} \bar{\partial}\omega \wedge \text{Tr} \left( -d_\Sigma \tilde{g} \tilde{g}^{-1} \wedge \mathcal{L}_B + \tilde{g} \mathcal{L}_A \tilde{g}^{-1} \wedge \mathcal{L}_B \right) ,$$

Upon using equation (2.14) and the third archipelago condition the final term of (5.20) reduces to:

$$\frac{i}{\hbar} \sum_{q \in P} \int_{\Sigma_q} \text{res}_q \left( \omega \wedge \left\langle -dg_q g_q^{-1}, \mathcal{L}_B \right\rangle + \omega \wedge \left\langle g_q \mathcal{L}_A g_q^{-1}, \mathcal{L}_B \right\rangle \right) \tag{5.21}$$

while the second and fourth terms give the unified sigma model action of [25], given in equation (3.33), thus the doubled action reduces to:

$$S_{\text{Dbld}}(A, B) = \frac{1}{2\pi\hbar} \int_{\Sigma \times \mathbb{C}P^1} \omega \wedge \text{Tr} \left( \mathcal{L}_A \wedge \bar{\partial} \mathcal{L}_A \right) - \frac{1}{2\pi\hbar} \int_{\Sigma \times \mathbb{C}P^1} \omega \wedge \text{Tr} \left( \mathcal{L}_B \wedge \bar{\partial} \mathcal{L}_B \right) \tag{5.22}$$

$$+ S_{\text{USM}}(g, \mathcal{L}_A) - \frac{i}{\hbar} \sum_{q \in P} \int_{\Sigma_q} \text{res}_q (\omega \wedge \left\langle -dg_q g_q^{-1}, \mathcal{L}_B \right\rangle + \omega \wedge \left\langle g_q \mathcal{L}_A g_q^{-1}, \mathcal{L}_B \right\rangle)$$

It is particularly important that the final condition in (4.31) (which in the chosen gauge imposes $\partial_z A_{\bar{z}} = 0$) holds when doing this. If our boundary condition were weaker we would not be able to impose the third archipelago conditions and would find additional terms in the above equation involving $z$ derivatives of $\tilde{g}$. The result would be an action which is not genuinely two-dimensional and is unlike any traditional integrable model.

Before we derive any sigma models from this action we first derive its equations of motion by performing the variation:

$$\mathcal{L}_A \longrightarrow \mathcal{L}'_A = \mathcal{L}_A + \epsilon l_A , \qquad \mathcal{L}'_B = \mathcal{L}_B \longrightarrow \mathcal{L}_B + \epsilon l_B , \qquad \tilde{g} \longrightarrow \tilde{g}' = \hat{g} e^{\epsilon \chi_g} , \tag{5.23}$$

under which the action transforms as $S_{\text{Dbld}} \longrightarrow S' = S_{\text{Dbld}} + \epsilon \delta S$, thus the equations of motion are found from:

$$\delta S = \frac{d}{d\epsilon} S' \bigg|_{\epsilon=0} = 0 . \tag{5.24}$$

This allows us to show that $\mathcal{L}_A$ and $\mathcal{L}_B$ are indeed the Lax connections which characterise the sigma model and that the boundary condition $A_\Sigma|_\mathbf{h} = B_\Sigma$ gives an additional equation of motion coupling the two together. Note, the variation of the unified sigma model action was calculated in [39] and is:

$$\delta S_{\text{USM}} = -\frac{i}{\hbar} \sum_{q \in P} \int_{\Sigma_q} \text{res}_q \left( \omega \wedge \left\langle l_A + [\chi_g, \mathcal{L}_A] - d_\Sigma \chi_g, g_q^{-1} dg_q \right\rangle + \omega \wedge \left\langle \mathcal{L}_A, d\chi_g \right\rangle \right) \tag{5.25}$$

Before we perform the above variation we first note that our boundary conditions, which we collectively denote by $\Omega_{bc}$, constrain $l_A$, $l_B$ and $\chi_g$. If $A$ and $B$, given by (5.18), transform to $A'$ and $B'$ then at the poles of $\omega$, these constraints are:

$$\frac{dA'}{d\epsilon} \bigg|_{\epsilon=0} = \tilde{g}(-d\chi_g + [\chi_g, \mathcal{L}_A] + l_A)\tilde{g}^{-1} = \lambda \in \Omega_{bc} , \qquad \frac{dB'}{d\epsilon} \bigg|_{\epsilon=0} = l_B \in \Omega_{bc} , \tag{5.26}$$

where the boundary condition $A_\Sigma|_\mathbf{h} = B_\Sigma$ implies $\lambda|_\mathbf{h} = l_B$.

The variation of the first and second terms of (5.22) is:

$$\delta I_1 + \delta I_2 = \left.\frac{d(I_1' + I_2')}{d\epsilon}\right|_{\epsilon=0} = \frac{1}{\pi\hbar}\int_{\Sigma\times\mathbb{CP}^1} \omega \wedge \left[\langle l_A, \bar\partial\mathcal{L}_A\rangle - \langle l_B, \bar\partial\mathcal{L}_B\rangle\right] \tag{5.27}$$

$$+ \frac{i}{\hbar}\sum_{q\in P}\int_{\Sigma_q} \text{res}_q\left(\omega \wedge \left[\langle l_A, \mathcal{L}_A\rangle - \langle l_B, \mathcal{L}_B\rangle\right]\right),$$

while the variation of the final term is:

$$\delta I_4 = \left.\frac{dI_4'}{d\epsilon}\right|_{\epsilon=0} = -\frac{i}{\hbar}\sum_{q\in P}\int_{\Sigma_q}\text{res}_q\left(\omega \wedge \left\langle g_q(-d_\Sigma\chi_g + [\chi_g, \mathcal{L}_A] + l_A)g_q^{-1}, \mathcal{L}_B\right\rangle\right) \tag{5.28}$$

$$+ \frac{i}{\hbar}\sum_{q\in P}\int_{\Sigma_q}\text{res}_q\left(\omega \wedge \left\langle l_B, (g_q d_\Sigma g_q^{-1} + g_q\mathcal{L}_A g_q^{-1})\right\rangle\right). \tag{5.29}$$

If we use the first of equations (5.26) the sum of these terms can be rewritten as:

$$\delta S = \frac{1}{\pi\hbar}\int_{\Sigma\times\mathbb{CP}^1}\omega \wedge \langle l_A, \bar\partial\mathcal{L}_A\rangle - \langle l_B, \bar\partial\mathcal{L}_B\rangle - \frac{i}{\hbar}\sum_{q\in P}\int_{\Sigma_q}\text{res}_q\left(\omega \wedge \left[\langle \lambda, d_\Sigma g_q g_q^{-1}\rangle + \langle\mathcal{L}_A, d_\Sigma\chi_g\rangle\right]\right)$$

$$+ \frac{i}{\hbar}\sum_{q\in P}\int_{\Sigma_q}\text{res}_q\left(\omega \wedge \left[\langle g_q^{-1}\lambda g_q + d_\Sigma\chi_g - [\chi_g, \mathcal{L}_A], \mathcal{L}_A\rangle - \langle l_B, \mathcal{L}_B\rangle\right]\right)$$

$$- \frac{i}{\hbar}\sum_{q\in P}\int_{\Sigma_q}\text{res}_q\left(\omega \wedge \left[\langle\lambda, \mathcal{L}_B\rangle - \langle l_B, g_q d_\Sigma g_q^{-1} + g_q\mathcal{L}_A g_q^{-1}\rangle\right]\right) = 0. \tag{5.30}$$

If we integrate by parts the terms involving $\mathcal{L}_A \wedge d_\Sigma\chi_g$ and use $\langle[\chi_g, \mathcal{L}_A], \mathcal{L}_A\rangle = 2\langle\chi_g\mathcal{L}_A, \mathcal{L}_A\rangle$ then this reduces to:

$$\delta S = \frac{1}{\pi\hbar}\int_{\Sigma\times\mathbb{CP}^1}\omega \wedge \left[\langle l_A, \bar\partial\mathcal{L}_A\rangle - \langle l_B, \bar\partial\mathcal{L}_B\rangle\right] - \frac{2i}{\hbar}\sum_{q\in P}\int_{\Sigma_q}\text{res}_q\left(\omega \wedge \langle\chi_g, d_\Sigma\mathcal{L}_A + \mathcal{L}_A \wedge \mathcal{L}_A\rangle\right)$$

$$+ \frac{i}{\hbar}\sum_{q\in P}\int_{\Sigma_q}\text{res}_q\left(\omega \wedge \left\langle\lambda + l_B, g_q d_\Sigma g_q^{-1} + g_q\mathcal{L}_A g_q^{-1} - \mathcal{L}_B\right\rangle\right) = 0. \tag{5.31}$$

Thus our equations of motion are:

$$\omega \wedge \bar\partial\mathcal{L}_A = 0, \qquad \omega \wedge \bar\partial\mathcal{L}_B = 0, \qquad (d_\Sigma\mathcal{L}_A + \mathcal{L}_A \wedge \mathcal{L}_A)|_{(q,\bar q)} = 0 \tag{5.32}$$

$$(g_q d_\Sigma g_q^{-1} + g_q\mathcal{L}_A g_q^{-1})|_\mathbf{h} = \mathcal{L}_B, \tag{5.33}$$

where the third and final equations plus our boundary conditions imply the flatness of $\mathcal{L}_B$. These are of course in agreement with those derived above using the doubled equations of motion. In the following, we use equation (3.20), which is a solution to the first and second equations, to derive the Lax connection. Thus, we impose the first two equations on the action (5.22) and find the unified gauged sigma model action:

$$S_{\text{UGSM}} \equiv S_{\text{USM}}(g, \mathcal{L}_A) - \frac{i}{\hbar}\sum_{q\in P}\int_{\Sigma_q}\text{res}_q(\omega \wedge \left\langle -d_\Sigma g_q g_q^{-1}, \mathcal{L}_B\right\rangle + \omega \wedge \left\langle g_q\mathcal{L}_A g_q^{-1}, \mathcal{L}_B\right\rangle) \tag{5.34}$$

## 5.5 Example

In this following section we use the boundary conditions of section 4.1 along with generic form of the Lax connection (3.20) and the unified gauged sigma model action (5.34) to derive various sigma models. The form of the Lax connections at a pole $q$ of $\omega$ is given by (5.35),

$$\mathcal{L}_A|_{(q,\bar{q})} = g_q^{-1}A|_{(q,\bar{q})}g_q + g_q^{-1}dg_q, \qquad \mathcal{L}_B|_{(q,\bar{q})} = B|_{(q,\bar{q})} \tag{5.35}$$

We always assume that $\omega$ has a pole at infinity at which $g_\infty = 1$. For ease, in the following examples we fix $\Sigma = \mathbb{R}^2$ with Lorentzian signature and light-cone coordinates $x^\pm$.

### 5.5.1 The Gauged WZW Model

We consider the four-dimensional Chern-Simons action where $\omega$ is:

$$\omega = \frac{z - k}{z}dz, \tag{5.36}$$

with a zero at $z = k$, a simple pole at $z = 0$ and a double pole at $z = \infty$. At $z = 0$ we impose the gauged chiral boundary condition:

$$A_- = B_-, \qquad A_+|_{\mathbf{h}} = B_+, \tag{5.37}$$

while at $z = \infty$ we impose the gauged Dirichlet boundary condition:

$$A_\pm|_{\mathbf{f}} = 0, \qquad A_\pm|_{\mathbf{h}} = B_\pm, \qquad \partial_z A_\pm|_{\mathbf{h}} = \partial_z B_\pm. \tag{5.38}$$

We also choose $\tilde{g}$ such that $\tilde{g}_\infty = g_\infty = 1$ and denote $\tilde{g}_0 = g_0 = g$.

These boundary conditions constrain the pole of $\mathcal{L}_A$ allowed at the zero $z = k$ to be in $\mathbf{f}_\mathbb{C}$ which we show by considering the limit $k \to \infty$. Working in the inverse coordinates $z = 1/w$ it is clear that:

$$\lim_{k \to \infty} \frac{kw - 1}{w^2}dw = k\frac{dw}{w}, \tag{5.39}$$

where the zero at $k$ cancels the double pole at infinity, leaving a simple pole while removing the zero from the action. In the four-dimensional Chern-Simons action this limit leaves an overall factor of $k$ which can be absorbed into $\hbar$.

Since this limit is permissible within the action we must choose a setup of defects (i.e. boundary conditions on, and poles of, $A$ and $B$) which leave a solution to the equations of motion after the limit is taken. As the described limit takes a pole of $A$ or $B$ at $k$ to $\infty$ and removes the zero from $\omega$ it follows that the resulting field configuration can no longer have a pole. Thus, our pole must cancel with the gauged Dirichlet boundary condition leaving only a gauged chiral or anti-chiral boundary condition (as $\omega$ has a simple pole at infinity after the limit). In all of our boundary conditions we have $A_\pm|_{\mathbf{h}} = B_\pm$ meaning there is no zero of $A_\pm|_{\mathbf{h}}$ or $B_\pm$ to cancel a pole, thus any poles of $A$ must be in $\mathbf{f}_\mathbb{C}$ where cancellation is possible due to zeros of $A_\pm|_{\mathbf{f}}$.

In this example, we demand the limit $k \to \infty$ reduces the gauged Dirichlet condition to the gauged anti-chiral boundary condition, thus imposing that $(z - k)A_-|_{\mathbf{f}}$ be regular. If we take $k$ to be outside the disc $U_0$, then the archipelago conditions mean $\tilde{g} = 1$ at $k$, hence by (5.35) we require that:

$$(z - k)\mathcal{L}_{A-}|_{\mathbf{f}} \qquad \text{is regular.} \tag{5.40}$$

Since $A_+$ and $B_\pm$ are regular at $k$ by the above argument, it follows from (5.35) and the solution to the equations of motion in the gauge $A_{\bar{z}} = 0$, equation (3.20), that $\mathcal{L}_A$ and $\mathcal{L}_B$ are of the form:

$$\mathcal{L}_A = \mathcal{L}_{A+}^c dx^+ + \left(\mathcal{L}_{A-}^c + \frac{\mathcal{L}_{A-}^{k,0}}{z - k}\right)dx^-, \qquad \mathcal{L}_{B\pm} = \mathcal{L}_{B\pm}^c = B_\pm(x^+, x^-), \tag{5.41}$$

where (5.40) means $\mathcal{L}_{-}^{k,0} \in \mathbf{f}_{\mathbb{C}}$. We note the final equality follows from the fact that $\mathcal{L}_B$ has no dependence upon $z$ and that $B = \mathcal{L}_B$.

Using (5.35), the boundary condition at $z = \infty$, equation (5.38), and $g_{\infty} = 1$ it follows that:

$$\mathcal{L}_{A\pm}^{c} = B_{\pm}\,. \tag{5.42}$$

Similarly, the boundary condition at $z = 0$, (5.37), with (5.35) and $g_0 = g$ implies:

$$\mathcal{L}_{A-}^{k,0} = k\left(B_{-} - g^{-1}\partial_{-}g - g^{-1}B_{-}g\right)\,, \quad \mathcal{L}_{A+}^{c} = g^{-1}\partial_{+}g + g^{-1}(A_{+}|_{\mathbf{f}} + B_{+})g\,, \tag{5.43}$$

Hence, the Lax connections are:

$$\mathcal{L}_A = B_{+}dx^{+} + \frac{1}{z-k}(zB_{-} - k(g^{-1}\partial_{-}g + g^{-1}B_{-}g))dx^{-}\,, \qquad \mathcal{L}_B = B_{+}d^{+} + B_{-}dx^{-}\,. \tag{5.44}$$

We also note that the condition $\mathcal{L}_{A-}^{k,0} \in \mathbf{f}_{\mathbb{C}}$ implies:

$$(g^{-1}\partial_{-}g + g^{-1}B_{-}g)|_{\mathbf{h}} = B_{-}\,, \tag{5.45}$$

while (5.42) and the second of equations (5.43) imply:

$$(g\partial_{+}g^{-1} + gB_{+}g^{-1})|_{\mathbf{h}} = B_{+}\,, \tag{5.46}$$

where we have conjugated the second equation in (5.43) by $g$ before projecting into $\mathbf{h}_{\mathbb{C}}$.

Having found the lax connections (5.44) we substitute them into the unified gauged sigma model action (5.34). Note, since $g_{\infty} = 1$ and thus $dg_{\infty} = 0$ we need only calculate $\text{res}_0(\omega \wedge \mathcal{L}_A)$:

$$\text{res}_0(\omega \wedge \mathcal{L}_A) = -kB_{+}dx^{+} - k(g^{-1}\partial_{-}g + g^{-1}B_{-}g)dx^{-}\,. \tag{5.47}$$

Hence, the unified sigma model term of (5.34) is:

$$S_{\text{USM}}(\mathcal{L}_A, \tilde{g}) = -\frac{ik}{\hbar}\int_{\mathbb{R}_0^2} d^2x\, \text{Tr}\left(g^{-1}\partial_{+}g g^{-1}\partial_{-}g + \partial_{+}g g^{-1}B_{-} - B_{+}g^{-1}\partial_{-}g\right) - \frac{ik}{3\hbar}\int_{\mathbb{R}^2 \times [0,R_0]} \text{Tr}(\tilde{g}^{-1}d\tilde{g})^3\,, \tag{5.48}$$

where $d^2x = dx^{+} \wedge dx^{-}$ while the Wess-Zumino term at $z = \infty$ vanishes since $\tilde{g} = 1$ at both $r_{\infty} = 0$ and $r_{\infty} = R_{\infty}$. Similarly, the second term in (5.34) only has contributions at $z = 0$ since $g_{\infty} = 1$, hence:

$$\frac{i}{\hbar}\sum_{q \in P}\int_{\mathbb{R}_q^2} \text{Tr}\left(g_q dg_q^{-1} \wedge \text{res}_q(\omega \wedge \mathcal{L}_B)\right) = \frac{i}{\hbar}\int_{\mathbb{R}_0^2} \text{Tr}(-dgg^{-1} \wedge \text{res}_0(\omega \wedge \mathcal{L}_B)) \tag{5.49}$$

$$= \frac{ik}{\hbar}\int_{\mathbb{R}_0^2} dx^{+} \wedge dx^{-}\, \text{Tr}(\partial_{+}gg^{-1}B_{-} - \partial_{-}gg^{-1}B_{+})\,,$$

while the final term gives:

$$\frac{i}{\hbar}\sum_{q \in P}\int_{\mathbb{R}_q^2} \text{Tr}\left(\text{res}_q(\omega \wedge g_q \mathcal{L}_A g_q^{-1} \wedge \mathcal{L}_B)\right) = \frac{ik}{\hbar}\int_{\mathbb{R}_0^2} dx^{+} \wedge dx^{-}\, \text{Tr}(\partial_{-}gg^{-1}B_{+} - gB_{+}g^{-1}B_{-} + B_{-}B_{+})$$

$$+ \frac{ik}{\hbar}\int_{\mathbb{R}_\infty^2} dx^{+} \wedge dx^{-}\, \text{Tr}(-g^{-1}\partial_{-}gB_{+} - g^{-1}B_{-}gB_{+} + B_{+}B_{-})\,. \tag{5.50}$$

Upon combining these three equations and setting $i\hbar = 4\pi$, we find the gauged WZW model action [32, 34]:

$$S_{\text{GWZW}}(g, B_{+}, B_{-}) = S_{\text{WZW}}(g) + \frac{k}{2\pi}\int_{\mathbb{R}^2} dx^{+} \wedge dx^{-}\, \text{Tr}(\partial_{+}gg^{-1}B_{-} - B_{+}g^{-1}\partial_{-}g - gB_{+}g^{-1}B_{-} + B_{+}B_{-})\,, \tag{5.51}$$

where $S_{\text{WZW}}(g)$ is the Wess-Zumino-Witten model. It is simple to demonstrate that the equations of motion of this model are the flatness of (5.44) at $z = 0$ as well as equations (5.45) and (5.46). This is in agreement with what one expects from the equations of motion of the unified gauged sigma model (5.32) and (5.33).

### 5.5.2   The Nilpotent Gauged WZW Model

In [38, 45], Balog et al. demonstrated the conformal Toda field theories and W-algebras can be found by constraining a version of the gauged WZW model; we call this version the nilpotent gauged WZW model. As we have discussed above, the Wess-Zumino-Witten model has the symmetry group, $G_L \times G_R$ where the $G_L$ acts from the left $g \to ug$ and is a function of $x^+$, $u(x^+)$, while the second acts on the right $g \to g\bar{u}$ and depends on $x^-$. What makes this version of the gauged WZW model unusual is that one gauges these two symmetries independently from each other, finding a model whose target space is $G/(N^- \times N^+)$. By introducing a gauge field $C$ we gauge the left symmetry by a maximal nilpotent subgroup of $G$ associated to positive roots, denoted by $N^+$; this field is valued in the Lie algebra $\mathbf{n}^+$ of $N^+$. Similarly, we introduce the gauge field $B$ to gauge the right symmetry by a maximal nilpotent subgroup of $G$ associated to negative roots, denoted by $N^-$, this field is valued in the Lie algebra $\mathbf{n}^-$ of $N^-$. We note $\mathbf{n}_{\mathbb{C}}^-, \mathbf{n}_{\mathbb{C}}^+ \subset \mathbf{g}_{\mathbb{C}}$. One recovers the Toda theories from the nilpotent gauged WZW model by fixing the gauge $C_- = B_+ = 0$ and performing a Gauss decomposition, as discussed in [38]. In this section we will assume $G_{\mathbb{C}} = SL(N, \mathbb{C})$ in which case we work in a basis where $\mathbf{n}_{\mathbb{C}}^+$ is the set of strictly upper triangular matrices, and $\mathbf{n}_{\mathbb{C}}^-$ is the set of strictly lower triangular matrices. The case of $G_{\mathbb{C}}$ is recovered by replacing $\mathbf{n}_{\mathbb{C}}^+$ and $\mathbf{n}_{\mathbb{C}}^-$ by a pair maximal nilpotent subalgebras associated to positive and negative roots.

Consider a tripled version of the four-dimensional Chern-Simons model with three gauge fields $A \in \mathbf{sl}_{\mathbb{C}}(n)$, $B \in \mathbf{n}_{\mathbb{C}}^-$, $C \in \mathbf{n}_{\mathbb{C}}^+$:

$$S_{\text{Tripled}}(A, B, C) = S_{\text{4dCS}}(A) - S_{\text{4dCS}}(B) - S_{\text{4dCS}}(C) \tag{5.52}$$

$$-\frac{i}{\hbar} \int_{\mathbb{R}_0^2} \text{res}_0 \left( \langle \omega \wedge A, C \rangle + 2 \langle \omega \wedge A_- dx^-, \mu dx^+ \rangle \right) - \frac{i}{\hbar} \int_{\mathbb{R}_\infty^2} \text{res}_\infty \left( \langle \omega \wedge A, B \rangle + 2 \langle \omega \wedge A_+ dx^+, \nu dx^- \rangle \right),$$

where:

$$\omega = \frac{(z - z_-)}{z} dz, \tag{5.53}$$

while $\mu \in \mathbf{n}_{\mathbb{C}}^-$ and $\nu \in \mathbf{n}_{\mathbb{C}}^+$ are constants. We fix the manifold $\Sigma \times C$ to be $\mathbb{R}^2 \times \mathbb{C}P^1$ where $\mathbb{R}^2$ has the light-cone coordinates $x^\pm$ and metric $\eta^{+-} = 2, \eta_{++} = \eta_{--} = 0$. We take $A, B$ and $C$ to be in their respective adjoint representations.

For each of these algebras, as well as the Cartan subalgebra of $\mathbf{sl}_{\mathbb{C}}(n)$, denoted $\mathbf{g}_0$, we define our basis in the following way. For $\mathbf{n}_{\mathbb{C}}^+$ our basis is $\{e_\alpha\}$, for $\mathbf{n}_{\mathbb{C}}^-$ $\{e_{-\beta}\}$, for $\mathbf{g}_0$ $\{h_\gamma\}$, and for $\mathbf{sl}_{\mathbb{C}}(n)$ $\{h_\gamma, e_\alpha, e_{-\beta}\}$. The indices in each basis indicate that these elements are labelled by elements of root space of $\mathbf{sl}_{\mathbb{C}}$, denoted $\Phi$. The index $\gamma$ is in the set simple roots $\Delta$, while $\alpha$ and $\beta$ are positive roots in the space $\Phi^+$. In this basis the trace of $\mathbf{g}_{\mathbb{C}}$ is given by:

$$\langle e_\alpha, e_\beta \rangle = \frac{2}{\alpha^2} \delta_{\alpha, -\beta}, \qquad \langle h_\gamma, h_\tau \rangle = \gamma^\vee \cdot \tau^\vee, \qquad \langle e_\alpha, h_\gamma \rangle = 0, \tag{5.54}$$

where $\gamma, \tau \in \Delta$, $\alpha, \beta \in \Phi$, and $\alpha^\vee = 2\alpha/\alpha^2$ is the coroot [46, 47]. We have given the derivation of these traces in appendix E. If we expand the actions $S_{\text{4dCS}}(B)$ and $S_{\text{4dCS}}(C)$ into their Lie algebra components, it is clear that $S_{\text{4dCS}}(B) = S_{\text{4dCS}}(C) = 0$ by the first of equation in (5.54) where $\langle e_\alpha, e_\beta \rangle = 0$ since $\beta \neq -\alpha$ as the elements of $\mathbf{n}_{\mathbb{C}}^+$ are labelled by the positive roots $\Phi^+$ while the elements of $\mathbf{n}_{\mathbb{C}}^-$ are labelled by the negative roots $\Phi^-$. Hence the action (5.52) reduces to:

$$S_{\text{Tripled}}(A, B, C) = S_{\text{4dCS}}(A) - \frac{i}{\hbar} \int_{\mathbb{R}_0^2} \text{res}_0 \left( \langle \omega \wedge A, C \rangle + 2 \langle \omega \wedge A_- dx^-, \mu dx^+ \rangle \right) \tag{5.55}$$

$$- \frac{i}{\hbar} \int_{\mathbb{R}_\infty^2} \text{res}_\infty \left( \langle \omega \wedge A, B \rangle + 2 \langle \omega \wedge A_+ dx^+, \nu dx^- \rangle \right), \tag{5.56}$$

hence the fields $B$ and $C$ behave as Lagrange multipliers.

Since $B$ and $C$ only appear in boundary terms we have one bulk equation of motion:

$$\omega \wedge F(A) = 0 \,, \tag{5.57}$$

where $A$ is gauge equivalent to a Lax connection $\mathcal{L}_A$ by $A = \hat{g}d\hat{g}^{-1} + \hat{g}\mathcal{L}_A\hat{g}^{-1}$. We note that as above $\hat{g}$ is defined by $A_{\bar{z}} = \hat{g}\partial_{\bar{z}}\hat{g}^{-1}$. Since $B$ and $C$ do not have any equations of motion in the bulk we assume $\partial_{\bar{z}}B = \partial_{\bar{z}}C = 0$.

If we vary $A$, $B$ and $C$ together while using (2.12) and (2.14) we find the defect equations of motion:

$$\int_{\mathbb{R}^2_0} \left( \langle A - C, \delta A \rangle + \langle A, \delta C \rangle + 2 \left\langle \delta A_- dx^-, \mu dx^+ \right\rangle \right) = 0 \,, \tag{5.58}$$

$$\int_{\mathbb{R}^2_\infty} (k - \partial_z) \left( \langle A - B, \delta A \rangle + \langle A, \delta B \rangle + 2 \left\langle \delta A_+ dx^+, \nu dx^- \right\rangle \right) = 0 \,. \tag{5.59}$$

We solve these two equations by expanding the Lie algebra components into $\mathbf{g}_0, \mathbf{n}^+_\mathbb{C}, \mathbf{n}^-_\mathbb{C}$ and introducing nilpotent versions of gauged chiral and Dirichlet boundary conditions:

$$A^\alpha_- = C^\alpha_- \,, \quad A^{-\alpha}_- = A^\gamma_- = 0 \,, \quad A^{-\alpha}_+ = \mu^{-\alpha} \quad \text{at } \boldsymbol{z} = (0,0) \,, \tag{5.60}$$

$$A^{-\alpha}_+ = B^{-\alpha}_+ \,, \quad A^\alpha_+ = A^\gamma_+ = 0 \,, \quad A^\alpha_- = \nu^\alpha \quad \text{at } \boldsymbol{z} = (\infty,\infty) \,, \tag{5.61}$$

$$\partial_z(A^{-\alpha}_+ - B^{-\alpha}_+) = 0 \,, \quad \partial_z A^\alpha_+ = \partial_z A^\gamma_+ = 0 \,, \quad \partial_z A^\alpha_- = 0 \quad \text{at } \boldsymbol{z} = (\infty,\infty) \,, \tag{5.62}$$

where $\alpha \in \Phi^+$ and $\gamma \in \Delta$.

As has been discussed above, one can only recover a two dimensional sigma model from the four-dimensional Chern-Simons theory if the action is finite and therefore that the Lagrangian is regular in $z$ near poles of $\omega$. We needn't worry about the boundary terms of (5.56) as these are already finite, nor do we worry about the simple pole as this is finite by [39]. Hence, we analyse the behaviour of the action around the double pole at infinity. If we perform the inversion $z = 1/w$ and expand $S_{\mathrm{4dCS}}(A)$ into its Lie algebra components one finds:

$$S_{\mathrm{4dCS}}(A) = -\frac{1}{2\pi\hbar} \int_{\mathbb{R}^2 \times \mathbb{C}P^1} \frac{(kw-1)}{w^2} dw \wedge \left( \frac{2}{\alpha^2} \left( A^\alpha \wedge dA^{-\alpha} + A^{-\alpha} \wedge dA^\alpha \right) \right. \tag{5.63}$$

$$\left. + \gamma^\vee \cdot \tau^\vee A^\gamma \wedge dA^\tau - \frac{1}{3} \gamma^\vee \cdot \alpha^\vee A^\gamma \wedge A^\alpha \wedge A^{-\alpha} \right) \,,$$

where $\gamma, \tau \in \Delta$ and $\alpha \in \Phi^+$. Upon applying the boundary conditions (5.61) and using the argument of [39] to remove a power of $w$ we find the non-regular part of the Lagrangian density near $z = \infty$ is:

$$L(A) \sim \frac{1}{w} \left( \frac{2}{\alpha^2} \left[ \epsilon^{ij} A^\alpha_{\bar{w}} \partial_i A^{-\alpha}_j + \epsilon^{ij} A^{-\alpha}_i \partial_j A^\alpha_{\bar{w}} - \nu^\alpha \partial_+ A^{-\alpha}_{\bar{w}} \right] - \frac{1}{3} \gamma^\vee \cdot \alpha^\vee \left[ A^\gamma_- A^\alpha_{\bar{w}} B^{-\alpha}_+ - A^\gamma_{\bar{w}} \nu^\alpha B^{-\alpha}_+ \right] \right.$$

$$\left. - \gamma^\vee \cdot \tau^\vee \left[ A^\gamma_- \partial_+ A^\tau_{\bar{w}} - A^\gamma_{\bar{w}} \partial_+ A^\tau_- \right] \right) \tag{5.64}$$

where $\epsilon^{+-\bar{w}} = 1$ and $\epsilon^{+-} = 1$. Note, we have made use of the fact $\nu$ is constant and that $\partial_{\bar{z}}B = 0$. Clearly the Lagrangian density is only regular if $A_{\bar{w}} = 0$ at $w = 0$, or in the original coordinates $A_{\bar{z}} = 0$ at $z = \infty$.

In section 3 the condition $A_{\bar{z}} = 0$ at pole of $\omega$ was implemented via a gauge choice on $A$. In fact in section 3.2 we used the third archipelago condition to make this gauge choice by expressing the gauge field $A$ as $A = \tilde{g}d\tilde{g}^{-1} + \tilde{g}\mathcal{L}_A\tilde{g}^{-1}$, where $\tilde{g}$ satisfies the archipelago conditions. Whether we can do this depends on if we can construct $\tilde{g}$ from $\hat{g}$ by a gauge transformations of $A$ such that $\tilde{g} = u\hat{g}$. This requires that gauge transformations of $A$ by $u = \hat{g}\tilde{g}^{-1}$ preserve the boundary conditions on $A$ at poles of $\omega$. If we define $\tilde{g}$ as in

section 3.2. The boundary conditions (5.60) are preserved by the gauge transformation $A \to u(d + A)u^{-1}$ if $u$ is in the intersection of $N_{\mathbb{C}}^+$ and the centraliser of $\mu$. Since $u = \hat{g}\tilde{g}^{-1}$ is the identity at $z = 0$, which is contained in both of these groups, it follows that we can always perform the transformation $\hat{g} \to \tilde{g} = u\hat{g}$ for the boundary conditions in (5.60). Similarly, the boundary conditions (5.61,5.62) are preserved if $u$ is in the intersection of $N_{\mathbb{C}}^-$ and the centraliser of $\nu$. Both of these groups contain the identity, hence we can always perform the transformation $\hat{g} \to \tilde{g} = u\hat{g}$ for the boundary conditions in (5.61,5.62).

Since the boundary conditions (5.60,5.61,5.62) are preserved by the gauge transformation generated by $u = \hat{g}\tilde{g}^{-1}$, it follows that we can simplify the bulk action $S_{\text{4dCS}}(A)$ using the archipelago conditions, such that (5.56) becomes:

$$S_{\text{Tripled}}(A, B, C) = S_{\text{USM}}(\tilde{g}, \mathcal{L}_A) - \frac{i}{\hbar} \int_{\mathbb{R}_0^2} \text{res}_0 \left( \langle \omega \wedge A, C \rangle + 2 \langle \omega \wedge A_- dx^-, \mu dx^+ \rangle \right) \tag{5.65}$$

$$- \frac{i}{\hbar} \int_{\mathbb{R}_\infty^2} \text{res}_\infty \left( \langle \omega \wedge A, B \rangle + 2 \langle \omega \wedge A_+ dx^+, \nu dx^- \rangle \right) ,$$

where $S_{\text{USM}}(\tilde{g}, \mathcal{L}_A)$ is the unified sigma model (3.33).

As in section 5.5.1 the one-form (5.53) in the limit $k \to \infty$ has only simple poles at $z = 0, \infty$. For reasons similar to those in section 5.5.1 we wish to preserve the boundary condition $A_+^{-\alpha} = B_+^{-\alpha}$ and $A_+^{\gamma} = 0$ in the limit $k \to \infty$ thus we demand that $(z - k)A_-|_{\mathbf{n}^+}$ is regular which as above implies that $(z - k)\mathcal{L}_{A-}|_{\mathbf{n}^+}$ must regular. Hence, upon using (3.20) our Lax connection is for the form:

$$\mathcal{L}_A = \mathcal{L}_{A+}^c dx^+ + \left( \mathcal{L}_{A-}^c + \frac{\mathcal{L}_{A-}^{k,0}}{z - k} \right) dx^- , \tag{5.66}$$

where $\mathcal{L}_{A-}^{k,0} \in \mathbf{n}_{\mathbb{C}}^+$.

We now use:

$$A_i|_{(q,\bar{q})} = g_q \partial_i g_q^{-1} + g_q \mathcal{L}_{A\,i} g_q^{-1} , \tag{5.67}$$

where $i = \pm$ and the boundary conditions (5.60,5.61) to fix $\mathcal{L}_A$. As above we take $\tilde{g}$ to be of the form:

$$\tilde{g}|_{(0,0)} = g_0 = g , \qquad \tilde{g}|_{(\infty,\infty)} = g_\infty = 1 . \tag{5.68}$$

The boundary conditions at $z = \infty$ (5.61) along with (5.67) and $g_\infty = 1$ imply:

$$\mathcal{L}_{A+}^c = B_+ , \qquad \mathcal{L}_{A-}^c = \nu , \tag{5.69}$$

while the boundary conditions at $z = 0$, (5.67) and $g_0 = g$ imply:

$$\mathcal{L}_{A-}^{k,0} = k \left( \nu - g^{-1}C_-g - g^{-1}\partial_-g \right) . \tag{5.70}$$

Thus, the Lax connection is:

$$\mathcal{L}_A = B_+ dx^+ + \frac{1}{z - k} \left( z\nu - k \left( g^{-1}\partial_-g + g^{-1}C_-g \right) \right) . \tag{5.71}$$

Note, the boundary conditions at $z = 0$ imply the condition:

$$(g\partial_+g^{-1} + g^{-1}B_+g)|_{\mathbf{n}_{\mathbb{C}}^-} = \mu , \tag{5.72}$$

while the condition $\mathcal{L}_{A-}^{k,0} \in \mathbf{n}_{\mathbb{C}}^+$ implies:

$$(g^{-1}\partial_-g + g^{-1}C_-g)|_{\mathbf{n}_{\mathbb{C}}^+} = \nu . \tag{5.73}$$

We now show that substituting (5.71) into (5.65) gives the nilpotent gauged WZW model. The unified sigma model term of (5.65) has residues at both $z = 0$ and $\infty$, however we needn't calculate $\operatorname{res}_\infty(\omega \wedge \mathcal{L}_A)$ since $dg_\infty = 0$ as $g_\infty = 1$ meaning there is no contribution from the pole at $\infty$. Thus, we calculate $\operatorname{res}_0(\omega \wedge \mathcal{L}_A)$ where we find:

$$\operatorname{res}_0(\omega \wedge \mathcal{L}_A) = -kB_+ dx^+ - k(g^{-1}\partial_- g + g^{-1}C_- g)dx^- , \tag{5.74}$$

thus the kinetic term of the unified sigma model is:

$$-\frac{i}{\hbar} \sum_{q \in \{0,\infty\}} \int_{\Sigma_q} \left\langle \operatorname{res}_q(\omega \wedge \mathcal{L}_A), g_q^{-1} dg_q \right\rangle \tag{5.75}$$

$$= -\frac{ik}{\hbar} \int_{\mathbb{R}_0^2} dx^+ \wedge dx^- \left( -\left\langle B_+, g^{-1}\partial_- g \right\rangle + \left\langle g^{-1}\partial_- g, g^{-1}\partial_+ g \right\rangle + \left\langle C_-, \partial_+ g g^{-1} \right\rangle \right) ,$$

Similarly, the other two residues in (5.65) are:

$$-\frac{i}{\hbar} \int_{\mathbb{R}_0^2} \operatorname{res}_0 \left( \langle \omega \wedge A, C \rangle + 2 \left\langle \omega \wedge A_- dx^-, \mu dx^+ \right\rangle \right) \tag{5.76}$$

$$= -\frac{ik}{\hbar} \int_{\mathbb{R}_0^2} dx^+ \wedge dx^- \left( \left\langle \partial_+ g g^{-1}, C_- \right\rangle - \left\langle g B_+ g^{-1}, C_- \right\rangle + 2 \left\langle C_-, \mu \right\rangle \right) ,$$

$$-\frac{i}{\hbar} \int_{\mathbb{R}_\infty^2} \operatorname{res}_\infty \left( \langle \omega \wedge A, B \rangle + 2 \left\langle \omega \wedge A_+ dx^+, \nu dx^- \right\rangle \right) \tag{5.77}$$

$$= -\frac{ik}{\hbar} \int_{\mathbb{R}_\infty^2} dx^+ \wedge dx^- \left( -\left\langle g^{-1}\partial_- g, B_+ \right\rangle - \left\langle g^{-1}C_- g, B_+ \right\rangle + 2 \left\langle B_+, \nu \right\rangle \right) ,$$

where we have used $\operatorname{Tr}(C_+C_-) = \operatorname{Tr}(B_+B_-) = 0$ since $\mathbf{n}_\mathbb{C}^+$ contains upper triangular matrices, and $\mathbf{n}_\mathbb{C}^-$ lower triangular matrices, only. Upon combining all of this together and setting $i\hbar = 4\pi$ we find the nilpotent gauged WZW model [38]:

$$S_{\text{Nilpotent}}(g, B_+, C_-) = S_{\text{WZW}}(g) + \frac{k}{2\pi} \int_{\mathbb{R}^2} d^2x \left( \left\langle \partial_+ g g^{-1}, C_- \right\rangle - \left\langle B_+, g^{-1}\partial_- g \right\rangle \right. \tag{5.78}$$

$$\left. - \left\langle g B_+ g^{-1}, C_- \right\rangle + \langle \mu, C_- \rangle + \langle \nu, B_+ \rangle \right) ,$$

where $S_{\text{WZW}}(g)$ is the WZW model and $d^2x = dx^+ \wedge dx^-$. When one varies the fields of this action one finds that the equations of motion are the requirement that the Lax connection (5.71) is flat at $z = 0$ and the constraints (5.72,5.73). It is known from [38] that one can classically find the Toda theories from this action. In this discussion we assumed $G_\mathbb{C} = SL(N, \mathbb{C})$ – one easily recovers the case of an arbitrary $G_\mathbb{C}$ by replacing $\mathbf{n}_\mathbb{C}^+$ and $\mathbf{n}_\mathbb{C}^-$ with the maximal nilpotent subalgebras associated to positive and negative roots.

# 6 Conclusion

We have reviewed the recent work of Costello and Yamazaki [12], and Delduc et al [25]. In these papers it was shown that one could solve the equations of motion of four-dimensional Chern-Simons theory (with two-dimensional defects inserted into the bulk) by defining a class of group elements $\{\hat{g}\}$ in terms of $A_{\bar{z}}$. Given a solution to the equations of motion, one finds an integrable sigma model by substituting the solution back into the four-dimensional Chern-Simons action. These sigma models are classical field theories on the defects inserted in to the four-dimensional Chern-Simons theory. In [25] it was shown the equivalence class

of Lax connections of an integrable sigma model are the gauge invariant content of $A$, where $\mathcal{L}$ is found from $A$ by preforming the Lax gauge transformation (3.7). That $\mathcal{L}$ satisfies the conditions of a Lax connection was due to the Wilson lines and bulk equations of motion of $A$.

In section 4 we introduced the doubled four-dimensional Chern-Simons theory, inspired by an analogous construction in three-dimensional Chern-Simons [29]. In this section we coupled together two four-dimensional Chern-Simons theory fields, where the second field was valued in a subgroup of the first, by introducing a boundary term. This boundary term had the effect of modifying the defect equations of motion enabling the introduction of new classes of gauged defects associated to the poles of $\omega$. In the rest of this section it was shown that the properties of four-dimensional Chern-Simons theory, such as its semi-topological nature or the unusual gauge transformation, are also present in the doubled theory, even with the introduction of the boundary term.

In section 5 we used the techniques of Delduc et al in [25] to derive the unified gauged sigma model action (3.33). It was found that this model is associated to two Lax connections, one each for $A$ and $B$, and some boundary conditions associated to the defects inserted in the bulk of the doubled theory. The unified gauged sigma model's equations of motion are the flatness of the Lax connections and the boundary conditions associated to the defects. We concluded in sectio 5.5.2 by deriving the Gauged WZW and Nilpotent Gauged WZW models, from which one finds the conformal Toda field theories [38].

Before we finish we wish to make some additional comments. The first of these is on the relation between the doubled four-dimensional action (4.3) and its equivalent in three-dimensions:

$$S(A, B) = S_{\mathrm{CS}}(A) - S_{\mathrm{CS}}(B) - \frac{1}{2\pi} \int_M d \langle A, B \rangle \tag{6.1}$$

In [31] it was proven that the four-dimensional Chern-Simons action for $\omega = dz/z$ is $T$-dual to the three-dimensional Chern-Simons action. By Yamazaki's arguments it is clear that the boundary term of the doubled action (4.3) for $\omega = dz/z$ is $T$-dual to the boundary term of (6.1), hence (4.3) and (6.1) are $T$-dual. As a result, we expect that arguments analogous to those used in section 5 can be used to derive the gauged WZW model from (6.1). It is important to note that this is different to the derivation of the gauged WZW model from Chern-Simons theory given in [29]. This is because the introduction of the boundary term leads to a modification of the defect equations of motion and therefore the boundary conditions. This contrasts with the construction given in [29] where a Lagrange multiplier was used to impose the relevant boundary conditions.

In [25] the authors introduced the Manin pair $(\mathbf{d}_{\mathbb{C}}, \boldsymbol{l}_{\mathbb{C}})$ where $\mathbf{d}_{\mathbb{C}}$ is a Lie algebra with an isotropic subalgebra $\boldsymbol{l}_{\mathbb{C}}$. Note, here we mean isotropic in the same sense as [12, 25] where for $a, b \in \boldsymbol{l}_{\mathbb{C}}$ we have $\langle a, b \rangle = 0$. The Manin pair is used to solve the defect equations of motion (2.18) for a first order pole of $\omega$ by requiring that at the pole the gauge field $A$ is valued in the isotropic algebra $\boldsymbol{l}_{\mathbb{C}}$.

This brings us to our second comment. The boundary conditions we defined for the doubled four-dimensional Chern-Simons theory above are not unique, we can in fact define two further classes of boundary condition. The first of these is a gauged version of the Manin pair boundary conditions at a first order pole of $\omega$. If $D_{\mathbb{C}}$ contains a subgroup $H_{\mathbb{C}}$, where $\mathbf{h} \neq \boldsymbol{l}_{\mathbb{C}}$, we can introduce a second field $B$ with gauge group $H_{\mathbb{C}}$. Therefore the gauged Manin pair boundary conditions are given by requiring our gauge fields satisfy: $A_i|_{\mathbf{h}} = B_i$ in $\mathbf{h}_{\mathbb{C}}$ while in the orthogonal complement $\mathbf{f}_{\mathbb{C}}$ we restrict $A$ to be in the isotropic algebra, $A_i|_{\mathbf{f}} \in \boldsymbol{l}$.

In [10, 12, 25] the authors defined a boundary condition for a pair of poles of $\omega$, considering the case where the Lie algebra of the gauge group contains a Manin triple $(\mathbf{d}, \boldsymbol{l}_1, \boldsymbol{l}_2)$. In the Manin triple both $\boldsymbol{l}_1$ and $\boldsymbol{l}_2$ are isotropic subalgebras of $\mathbf{d}$ such that[5] $\mathbf{d} = \boldsymbol{l}_1 \dot{+} \boldsymbol{l}_2$. Given the Manin triple, one solves the defect equations of motion by imposing that $A$ is valued in the isotropic subalgebras of the Manin pairs $(\mathbf{d}, \boldsymbol{l}_1)$ and $(\mathbf{d}, \boldsymbol{l}_2)$ at either pole. When $D$ contains a subgroup $H$ one can define a gauged version of this boundary

---

[5]Here $\dot{+}$ denotes the direct sum as a vector space.

condition in the doubled theory. One does this by requiring $A_i|_\mathbf{h} = B_i$ at both poles, while restricting $A_i|_\mathbf{f}$ to be in $l_1$ or $l_2$ at either pole.

In [25], reality conditions were imposed upon the action such that it was real. This requirement meant that first order poles of $\omega$ must be considered in pairs such that they are either: (a) complex conjugates or (b) on the real line. It was suggested that for a fixed $\omega$ the models found by imposing Manin triple boundary conditions in case (a) should be Poisson-Lie $T$-dual to those found from case (b), where one has also imposed Manin triple boundary conditions. It is hoped that the same is true for the gauged Manin triple boundary conditions.

Finally, our hope is that one can find new integrable gauged sigma models using the construction defined in section 5. This being said, there are several other problems which we have not discussed in this paper, but which we plan to cover in the future. These include $\lambda$- [48, 49], $\eta$- [50, 51], and $\beta$-deformations [52–54], this is expected to be similar to [55] and [56–58]; the generation of affine Toda models from four-dimensional Chern-Simons theory; the generation of gauged sigma models associated to a higher genus choice of $C$, we expect this to be analogous to the discussion near the end of [12]; how to find a set of Poisson commuting charges from $\mathcal{L}_A$ and $\mathcal{L}_B$ such that $\mathcal{L}_A$ and $\mathcal{L}_B$ are Lax connections; related to this is, the connection between our construction of gauged sigma models and that given by Gaudin models, this is likely similar to [24]; the quantum theory of the doubled action, our hope is that it describes the quantum theory of the sigma models one can find classically; and finally whether the results of [22] can be repeated for the doubled action, enabling us to find higher dimensional integrable gauged sigma models.

# Acknowledgements

I would like to thank my supervisor Gérard Watts for proposing this problem and the support he has provided during our many discussions. I would also like to thank Ellie Harris and Rishi Mouland for our discussions; Nadav Drukker who kindly provided comments on a previous version of this manuscript; Benoit Vicedo for his comments; and finally the anonymous referee whose comments we feel have greatly improved the above work. This work was funded by the STFC grant ST/T000759/1.

# A  The Regularised Action and Localisation

Throughout this paper we consider several examples where $\omega$ has poles with multiplicity greater than one. In this situation it was shown in [39] that the action (or any integral of its form) contains terms which are singular at poles with large multiplicity. As above $\omega$ is given by:

$$\omega = \eta^1_\infty dz + \sum_{q \in P^{\text{fin}}} \sum_{l=0}^{n_q-1} \frac{\eta^l_q}{(z-q)^{l+1}} dz\,, \qquad \text{where} \qquad \eta^l_q = \text{res}_q\left((z-q)^l \varphi(z)\right)\,, \tag{A.1}$$

and $\eta^1_\infty = \text{res}_\infty(\omega/z)$. Let $\Lambda = f(z)d\bar{z} \wedge dx^+ \wedge dx^-$ where the function $f(z)$ is assumed to be meromorphic with poles only at the zeros of $\omega$. An example of such a 3-form is $\text{CS}(A)$ in the gauge $A_z = 0$. To identify the singular part of $\omega \wedge \Lambda$ we note that a pole of degree $l+1$ can be rewritten in terms of derivatives:

$$\frac{1}{(z-q)^{l+1}} = \frac{(-1)^l}{(l)!} \partial_z^l \left(\frac{1}{z-q}\right)\,. \tag{A.2}$$

Using Leibniz's rule we have:

$$(\partial_z^l (z-q)^{-1})\Lambda = \sum_{r=0}^l (-1)^{-r} \frac{l!}{r!(l-r)!} \partial_z^{l-r}\left((z-q)^{-1}\partial_z^r \Lambda\right)\,, \tag{A.3}$$

hence we find:

$$\omega \wedge \Lambda = \eta_\infty^1 z \partial \Lambda + \sum_{q \in P^{\text{fin}}} \sum_{l=0}^{n_q-1} \frac{\eta_q^l}{l!} \frac{dz}{z-q} \wedge \partial_z^l \Lambda + \partial \psi \,, \tag{A.4}$$

where:

$$\psi = -\eta_\infty^1 z \Lambda + \sum_{q \in P^{\text{sing}}} \sum_{l=1}^{n_q-1} \sum_{r=0}^{l-1} \frac{(-1)^{l-r}}{r!(l-r)!} \partial_z^{l-r-1} \left( \frac{\eta_q^l}{z-q} \partial_z^r \Lambda \right) \,. \tag{A.5}$$

Here $P^{\text{sing}} \subseteq P^{\text{fin}}$ is the set of poles whose multiplicity is greater than one; it is at these poles that $\psi$ is singular. Finally, it is simple to show the second term on the right-hand side of (A.4) is regular by changing coordinates to polar coordinates centred on $q$. Since $\Lambda$ contains $d\bar{z}$ it follows that $dz \wedge \partial_z^l \Lambda$ is proportional to $dz \wedge d\bar{z} = 2ir dr \wedge d\theta$ thus canceling the pole from $z - q = re^{i\theta}$. The same argument applies to the first term if one works in the coordinates $z = r^{-1}e^{-i\theta}$.

There are two ways in which the singular part $\partial \psi$ can be removed from the action ensuring it is regular. The first method is subtract $\partial \psi$ from the action as was used in [39]. The second, implicitly used in [12] as well as in the following, is to restrict the bundles we consider to those satisfying boundary conditions such that $\psi$ is regular. This ensures integrals of $\partial \psi$ vanish since $\mathbb{C}P^1$ is compact.

In the following we often find integrals of the form:

$$I = \int_{\Sigma \times \mathbb{C}P^1} \omega \wedge \bar{\partial} \xi = -\int_{\Sigma \times \mathbb{C}P^1} \bar{\partial} \left( \omega \wedge \xi \right) + \int_{\Sigma \times \mathbb{C}P^1} \bar{\partial} \omega \wedge \xi \,, \tag{A.6}$$

where $\xi$ is a 2-form on $\Sigma$. Using equations (A.2) and (A.3) as well as $\partial_{\bar{z}}(z-q)^{-1} = 2\pi i \delta^2(z-q)$ we can evaluate the second integral in the final equality and find:

$$I = 2\pi i \eta_\infty^1 \int_{\Sigma_\infty} \partial_z \xi + 2\pi i \sum_{q \in P^{\text{fin}}} \sum_{l=0}^{n_q-1} \frac{\eta_q^l}{l!} \int_{\Sigma_q} \partial_z^l \xi - \int_{\Sigma \times \mathbb{C}P^1} \bar{\partial} \left( \omega \wedge \xi \right) + \int_{\Sigma \times \mathbb{C}P^1} \partial \tilde{\psi} \,, \tag{A.7}$$

where $\Sigma_q = \Sigma \times (q, \bar{q})$ and:

$$\tilde{\psi} = 2\pi i \eta_\infty^1 \delta^2(1/z) d\bar{z} \wedge \xi - 2\pi i \sum_{q \in P^{\text{sing}}} \sum_{l=1}^{n_q-1} \sum_{r=0}^{l-1} \frac{(-1)^{l-r}}{r!(l-r)!} \eta_q^l \partial_z^{l-r-1} \left( \delta^2(z-q) d\bar{z} \wedge \partial_z^r \xi \right) \,. \tag{A.8}$$

Integrals of the form $\bar{\partial}(\omega \wedge \xi)$, with $\omega \wedge \xi$ meromorphic in $z$, can be sent to zero because $\mathbb{C}P^1$ is compact. When this is done (A.6) reduces to:

$$I = \int_{\Sigma \times \mathbb{C}P^1} \omega \wedge \bar{\partial} \xi = 2\pi i \eta_\infty^1 \int_{\Sigma_\infty} \partial_z \xi + 2\pi i \sum_{q \in P^{\text{fin}}} \sum_{l=0}^{n_q-1} \frac{\eta_q^l}{l!} \int_{\Sigma_q} \partial_z^l \xi \,. \tag{A.9}$$

Note, we can write and evaluate (A.9) as a sum over residues. Let $V_q \in \mathbb{C}P^1$ denote an open region which contains only the pole $q$, thus we have:

$$\int_{\Sigma_q} \text{res}_q \left( \omega \wedge \xi \right) = \int_{\Sigma \times V_q} d\bar{z} \wedge \delta^2(z-q) \partial_z^{n_q-1} \left( \frac{(z-q)^{n_q}}{(n_q-1)!} \omega \wedge \xi \right)$$

$$= \sum_{l=0}^{n_q-1} \int_{\Sigma \times V_q} d\bar{z} \wedge \delta^2(z-q) \binom{n_p-1}{l} \partial_z^{n_q-l-1} \left( \frac{(z-q)^{n_p-l}}{(n_p-1)!} (z-q)^l \omega \right) \wedge \partial_z^l \xi$$

$$= \sum_{l=0}^{n_q-1} \frac{\eta_q^l}{l!} \int_{\Sigma_q} \partial_z^l \xi \,, \tag{A.10}$$

where in the final equality we have cancelled $(n_q - 1)!$ with the same term in the binomial coefficient, used $\eta_q^l = \text{res}_q\left((z - q)^l \omega\right)$ and evaluated an integral over $V_q$.

# B Künneth Theorem and Cohomology

Künneth theorem gives one a relation between the cohomologies of a product space and the cohomologies of the manifolds which it is constructed from:

$$H^k(X \times Y) = \bigoplus_{i+j=k} H^i(X) \otimes H^j(Y). \tag{B.1}$$

The de Rham cohomology for $\mathbb{R}^n$ is:

$$H^k(\mathbb{R}^n) \cong \begin{cases} \mathbb{R}, & \text{if } k = 0, \\ 0, & \text{otherwise.} \end{cases} \tag{B.2}$$

While for $\mathbb{C}P^n$ this is:

$$H^k(\mathbb{C}P^n) \cong \begin{cases} \mathbb{R}, & \text{for k even and } 0 \leq k \leq 2n, \\ 0, & \text{otherwise.} \end{cases} \tag{B.3}$$

# C Unified Sigma Model Action Derivation

In this section we repeat the derivation of the Wess-Zumino term of unified sigma model (3.30) as given in [25]. We do this by using the first archipelago condition to localise to the discs $U_q$ of $\mathbb{C}P^1$ around poles in which $\tilde{g}$ is not the identity. Outside of these charts, $\tilde{g} = 1$ so these regions do not contribute to our integral. This leaves us with the equation:

$$\frac{1}{6\pi\hbar} \int_{\Sigma \times C} \omega \wedge \text{Tr}(\tilde{g}^{-1}d\tilde{g} \wedge \tilde{g}^{-1}d\tilde{g} \wedge \tilde{g}^{-1}d\tilde{g}) = \frac{1}{6\pi\hbar} \sum_{q \in P} \int_{\Sigma \times U_q} \omega \wedge \text{Tr}(\tilde{g}_q^{-1}d\tilde{g}_q \wedge \tilde{g}_q^{-1}d\tilde{g}_q \wedge \tilde{g}_q^{-1}d\tilde{g}_q). \tag{C.1}$$

One can simplify this equation further by using the second archipelago condition. In each disc $U_q$ centred on the pole $q$, we introduce polar coordinates around each pole, $z = q + r_q e^{i\theta_q}$, while if there is a pole at infinity we take $z = r_\infty^{-1} e^{-i\theta_\infty}$. The second archipelago condition means that only $d\theta_q$ contributes in $dz$[6], hence equation (C.1) becomes:

$$\frac{i}{6\pi\hbar} \sum_{q \in P \setminus \{\infty\}} \int_{\Sigma \times [0,R_q] \times [0,2\pi]} r_q \varphi(q + r_q e^{i\theta_q}) d\theta_q \wedge \text{Tr}(\tilde{g}_q^{-1}d\tilde{g}_q \wedge \tilde{g}_q^{-1}d\tilde{g}_q \wedge \tilde{g}_q^{-1}d\tilde{g}_q) \tag{C.2}$$

$$- \frac{i}{6\pi\hbar} \int_{\Sigma \times [0,R_\infty] \times [0,2\pi]} r_\infty \varphi(r_\infty^{-1} e^{-i\theta_\infty}) d\theta_\infty \wedge \text{Tr}(\tilde{g}_\infty^{-1}d\tilde{g}_\infty \wedge \tilde{g}_\infty^{-1}d\tilde{g}_\infty \wedge \tilde{g}_\infty^{-1}d\tilde{g}_\infty),$$

where $R_q$ is the radius of the disc $U_q$. Upon integrating over $\theta$ on each disc we find:

$$\frac{1}{6\pi\hbar} \int_{\Sigma \times C} \omega \wedge \text{Tr}(\tilde{g}^{-1}d\tilde{g} \wedge \tilde{g}^{-1}d\tilde{g} \wedge \tilde{g}^{-1}d\tilde{g}) \tag{C.3}$$

$$= \frac{i}{3\hbar} \sum_{q \in P} \text{res}_q(\omega) \int_{\Sigma \times [0,R_q]} \text{Tr}(\tilde{g}_q^{-1}d\tilde{g}_q \wedge \tilde{g}_q^{-1}d\tilde{g}_q \wedge \tilde{g}_q^{-1}d\tilde{g}_q).$$

---

[6]This is because $\partial_\theta \hat{g} = 0$ meaning $\text{Tr}(\tilde{g}_q^{-1}d\tilde{g}_q \wedge \tilde{g}_q^{-1}d\tilde{g}_q \wedge \tilde{g}_q^{-1}d\tilde{g}_q)$ is a three form of $dx^i \wedge dx^j \wedge dr$ where $i = \pm$.

# D  WZW and Gauged WZW Model Conventions

The WZW model is constructed from the field $g : \mathbb{R}^2 \to G$, where $G$ is a complex Lie group, and is defined by the action:

$$S_{\text{WZW}}(g) = \frac{k}{8\pi} \int_{\mathbb{R}^2} d^2x \sqrt{-\eta} \eta^{\mu\nu} \left\langle g^{-1}\partial_\mu g, g^{-1}\partial_\nu g \right\rangle + \frac{k}{12\pi} \int_B \left\langle g^{-1}dg, g^{-1}dg \wedge g^{-1}dg \right\rangle, \qquad \text{(D.1)}$$

where $d^2x = dx^+ \wedge dx^-$, $\eta^{\mu\nu}$ a metric on $\mathbb{R}^2$, $\eta$ the determinant of $\eta_{\mu\nu}$, and $\hat{g}$ the extension of $g$ into the three-dimensional manifold $B$, where $\partial B = \mathbb{R}^2$. In this paper we take $B = \mathbb{R}^2 \times [0, R_0]$ with light-cone coordinates $x^\pm$ on $\mathbb{R}^2$ and metric $\eta^{+-} = 2, \eta^{++} = \eta^{--} = 0$. Our light-cone coordinates are connected to the Lorentzian coordinates $x^0, x^1$ by $x^+ = x^0 + x^1$ and $x^- = x^0 - x^1$ with the Minkowski metric $\eta_{00} = -\eta_{11} = 1, \eta_{01} = 0$.

The WZW action is invariant under transformations of the form $g \to u(x^+) g \bar{u}(x^-)^{-1}$ in $G_L \times G_R$ where $u \in G_L$ and $\bar{u} \in G_R$. To show this invariance one defines an extension of $u$ and $\bar{u}$ into $B$, denoted $\hat{u}$, and uses the Polyakov-Wigmann identity:

$$S_{\text{WZW}}(gh) = S(g) + S(h) + \frac{k}{2\pi} \int_{\mathbb{R}^2} dx^+ \wedge dx^- \left\langle g^{-1}\partial_- g, \partial_+ h h^{-1} \right\rangle, \qquad \text{(D.2)}$$

to expand $S_{\text{WZW}}(u g \bar{u})$ into a sum over WZW terms. Upon doing this one finds all terms other than $S_{\text{WZW}}(g)$ vanish. On $B = \mathbb{R}^2 \times [0, R_0]$ we parametrise $[0, R_0]$ by $z$ and define the extension $\hat{u}$ such that $\hat{u}|_{z=0} = \bar{u}$ and $\hat{u}|_{z=R_0} = u$, this ensures a cancellation of the Wess-Zumino terms associated to $u$ and $\bar{u}$. All other terms vanish due to $\partial_- u = \partial_+ \bar{u} = 0$.

From the variation $g \to g + \delta g$ in (D.1) one finds the variation of the action:

$$\delta S(g) = -\frac{k}{2\pi} \int_{\mathbb{R}^2} dx^+ \wedge dx^- \left\langle g^{-1}\delta g, \partial_+(g^{-1}\partial_- g) \right\rangle = -\frac{k}{2\pi} \int_{\mathbb{R}^2} dx^+ \wedge dx^- \left\langle \delta g g^{-1}, \partial_-(\partial_+ g g^{-1}) \right\rangle, \qquad \text{(D.3)}$$

which gives the equations of motion:

$$\partial_+(g^{-1}\partial_- g) = \partial_-(\partial_+ g g^{-1}) = 0, \qquad \text{(D.4)}$$

where $J_+ = \partial_+ g g^{-1}$ and $J_- = g^{-1}\partial_- g$ are the currents of the model. These equations have the solution:

$$g(x^+, x^-) = g_l(x^+) g_r(x^-)^{-1}, \qquad \text{(D.5)}$$

where $g_l$ ($g_r$) is a generic holomorphic (anti-holomorphic) map into $G$.

One can define a version of the WZW model where the symmetry $g \to u g \bar{u}^{-1}$ is gauged by a group $H \subseteq G$, this gives an action to the coset models [26–28] as shown in [32–36]. This gauged WZW model can be found from the normal WZW model by applying the Polyakov-Wigmann identity (D.2) to:

$$S_{\text{Gauged}}(g, h, \tilde{h}) = S_{\text{WZW}}(h g \tilde{h}^{-1}) - S_{\text{WZW}}(h \tilde{h}^{-1}), \qquad \text{(D.6)}$$

where $h(x^+, x^-), \tilde{h}(x^+, x^-) \in H$. It is clear that this equation is invariant under the transformation $g \to u g u^{-1}, h \to h u^{-1}, \tilde{h} \to \tilde{h} u^{-1}$ for $u(x^+, x^-) \in H$. After expanding (D.6) and setting $B_- = h^{-1}\partial_- h$ and $B_+ = \tilde{h}^{-1}\partial_+ \tilde{h}$ one finds gauged WZW model action:

$$S_{\text{Gauged}}(g, B_+, B_-) = S_{\text{WZW}}(g) + \frac{k}{2\pi} \int_{\mathbb{R}^2} dx^+ \wedge dx^- \left( \left\langle \partial_+ g g^{-1}, B_- \right\rangle \right. \qquad \text{(D.7)}$$

$$\left. - \left\langle B_+, g^{-1}\partial_- g \right\rangle - \left\langle g B_+ g^{-1}, B_- \right\rangle + \left\langle B_+, B_- \right\rangle \right),$$

where the symmetry $g \to ugu^{-1}, h \to hu^{-1}, \tilde{h} \to \tilde{h}u^{-1}$ corresponds to the gauge transformation:

$$g \longrightarrow ugu^{-1}, \qquad B_\pm \longrightarrow u(\partial_\pm + B_\pm)u^{-1}, \tag{D.8}$$

for $u(x^+, x^-) \in H$. This gauge symmetry means the orbits of $G$ which are mapped to each other by the action of $H$ are identified and therefore physical equivalent, hence the target space of the gauged WZW model is the coset $G/H$.

It is important to note that two conventions for the WZW model and Polyakov-Wigmann identity exist which are related by $g \to g^{-1}, h \to h^{-1}$. Further still, four conventions for the gauged WZW models exist found by taking $g \to g^{-1}$ and $B_+ \to -B_+$ in various combinations.

# E  The Cartan-Weyl Basis

Here we collect some facts about Lie algebras from [43]. A semi-simple Lie algebra $\mathbf{g}$ can be decomposed into three subalgebras $\mathbf{g} = \mathbf{n}^+ \oplus \mathbf{g}_0 \oplus \mathbf{n}^-$. The first subalgebra is a Cartan subalgebra $\mathbf{g}_0$ which is a maximal set of commuting semi-simple[7] elements of $\mathbf{g}$. We take $\{H_i\}$ to be a basis of $\mathbf{g}_0$. We can choose a basis of $\mathbf{n}^+$ to be $\{e_\alpha\}$ where the set $\{\alpha\} = \Phi^+$ is called the set of positive roots. Similarly, $\{e_{-\alpha}\}$ span $\mathbf{n}^-$ and $\{-\alpha\} = \Phi^-$ is the set of negative roots.

The Killing form on $\mathbf{g}$ is $K(x, y) = \mathrm{Tr}(\mathrm{ad}_x \circ \mathrm{ad}_y)$, where $\mathrm{ad}_x$ denotes the adjoint action of $x$, $\circ$ the composition of maps, and Tr over linear maps. Let $\langle \cdot, \cdot \rangle$ denote a symmetric invariant bilinear form proportional to the Killing form. We can always choose the basis elements $H_i$ to be orthonormal. With these choices, we can take the commutators to be

$$[H_i, H_j] = 0, \qquad\qquad [H_i, e_{\pm\alpha}] = \pm\alpha^i e_{\pm\alpha}, \tag{E.1}$$

$$[e_\alpha, e_{-\alpha}] = \frac{2\alpha^i}{\alpha^2}H_i, \qquad\qquad [e_{\pm\alpha}, e_{\pm\beta}] = \epsilon(\pm\alpha, \pm\beta)e_{\pm\alpha\pm\beta}. \tag{E.2}$$

where $\epsilon(\pm\alpha, \pm\beta)$ is a structure constant for any pair of $\pm$, $\alpha^i$ the $i$-th element of $\alpha \in \Phi^+$, and $\alpha^2 = \langle\alpha, \alpha\rangle$. If in the final equation $\pm\alpha \pm \beta \notin \Phi^\pm$ then $\epsilon(\pm\alpha, \pm\beta) = 0$.

Let $\Delta$ denote the set of generators of $\Phi^+$, which are called simple roots. For each root $\alpha \in \Phi$ one can define an element of the Cartan Subalgebra given by $h_\alpha = \alpha_i^\vee H_i$, where $\alpha_i^\vee = 2\alpha_i/\alpha^2$ is the coroot. The set of elements $\{h_\alpha\}$ labelled by a simple root, $\alpha \in \Delta$, form a basis of the Cartan subalgebra. From this result the equations (E.1,E.2) can be rewritten as:

$$[h_\gamma, h_\tau] = 0, \qquad\qquad [h_\gamma, e_{\pm\beta}] = \pm\gamma^\vee \cdot \beta e_{\pm\beta}, \tag{E.3}$$

$$[e_\alpha, e_{-\alpha}] = h_\alpha, \qquad\qquad [e_{\pm\alpha}, e_{\pm\beta}] = \epsilon(\pm\alpha, \pm\beta)e_{\pm\alpha\pm\beta}, \tag{E.4}$$

where $\gamma, \tau \in \Delta$ and $\alpha, \beta \in \Phi^+$. Note, to each root $\alpha \in \Phi^+$ we can associate an $\mathbf{sl}_\mathbb{C}(2)$ given by $\mathbf{g}_\alpha = \{e_\alpha, e_{-\alpha}, h_\alpha\}$.

The inner product $\langle h_\alpha, h_\beta\rangle$ is found by noting the basis elements $\{H_i\}$ are orthonormal, i.e. $\langle H_i, H_j\rangle = \delta_{ij}$, hence:

$$\langle h_\alpha, h_\beta\rangle = \frac{4\alpha_i\beta_j}{\alpha^2\beta^2}\langle H_i, H_j\rangle = \langle\alpha^\vee, \beta^\vee\rangle, \tag{E.5}$$

where $\langle\alpha^\vee, \beta^\vee\rangle$ is the symmetrised Cartan matrix. The final bilinear form to be found is $\langle e_\alpha, e_{-\alpha}\rangle$. Using the identity $\langle X, [Y, Z]\rangle = \langle[X, Y], Z\rangle$ it is clear that:

$$\langle\alpha^\vee, \alpha\rangle \langle e_\alpha, e_{-\alpha}\rangle = \langle h_\alpha, [e_\alpha, e_{-\alpha}]\rangle = \langle h_\alpha, h_\alpha\rangle = \frac{4}{\alpha^2}, \tag{E.6}$$

---

[7]An element $x \in \mathbf{g}$ is semi-simple if the matrix of eigenvalues formed by the adjoint action $\mathrm{ad}_x$ is diagonalisable.

hence our trace in the basis $\{h_\gamma, e_\alpha, e_{-\beta}\}$ is:

$$\langle e_\alpha, e_\beta \rangle = \frac{2}{\alpha^2} \delta_{\alpha, -\beta} \,, \qquad \langle h_\gamma, h_\tau \rangle = \gamma^\vee \cdot \tau^\vee \,, \qquad \langle e_\alpha, h_\gamma \rangle = 0 \,, \tag{E.7}$$

where $\gamma, \tau \in \Delta$ and $\alpha, \beta \in \Phi$.

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
