# Peer review of "Four-Dimensional Chern-Simons and Gauged Sigma Models"

_SciPost Physics Core_

## Round 5 · Referee Report · Anonymous · 2024-2-13

Report

This is a revised version of the manuscript, which deals with the construction of integrable gauged sigma-models from a doubled 4d Chern-Simons theory.

In my opinion, the changes implemented by the author have improved the paper and partially addressed some of the important problems contained in the previous version. However, I think there are still issues and mistakes in the revised manuscript, which I will explain below. My report on the previous version was followed by an extensive exchange with the author, accessible under the following link:
https://scipost.org/submissions/2109.08101v4/#report_1
Each step in this exchange came with new errors, making the reviewing process very slow and difficult. For this reason and seeing that the new version still contains mistakes, it seems to me that the submission is not in a stage which will converge to a correct publishable version in a reasonable time frame. Thus, my recommendation is to reject this submission.

Below is a detailed explanation of the most important problems found in the new version.

1) In my opinion, the main issue is in the formulation of the weak archipelago condition in subsection 5.3. There are more precisely two such formulations, one in terms of a field $\tilde{g}$ valued in the group $G$ and one in terms of a field $\tilde{\jmath}$ valued in the jet group $J$ of $G$. To the best of my understanding, the second one is correct (and, as claimed, follows from the results of the reference [41]). However, the author states that these conditions can be equivalently rewritten in terms of $\tilde{g}$ and uses this reformulation to derive the 2d gauged sigma-models, which I think is not correct. More precisely, around a higher order pole $q$, I think it is not possible for the field $\tilde{g}$ to depend only on the radial coordinate $r=|z-q|$, as claimed in the condition (ii), page 30. Indeed, working in polar coordinates $z-q=r \, e^{i\theta}$, a direct computation would then give
$\tilde{g}^{-1}\partial_z \tilde{g} = \frac{1}{2} e^{-i\theta} \tilde{g}(r)^{-1}\tilde{g}'(r)$, such that the field $\Delta_q = \tilde{g}^{-1}\partial_z \tilde{g}|_{z=q}$ would not be well defined as it would depend on the angle $\theta$ with which we approach $z=q$ (contradicting the condition (iii) on page 30). Essentially, the issue is that the existence of a $J$-valued field $\tilde{\jmath}$ which depends only on $r$ does not mean that $\tilde{\jmath}$ is the jet of a $G$-valued field $\tilde{g}$ depending itself only on $r$.

Since this weak archipelago condition on $\tilde{g}$ is used to derive the main result of the paper, namely the unified action of the gauged sigma-model, this poses a serious problem. (The final claimed result might be correct but it is important to derive it without any errors or potential confusions, especially at this stage in the reviewing process)

2) The second main issue is with the computation of the Wess-Zumino term (5.21), done in appendix B. Beyond the potential problems that arise from the non-existence of a radial $\tilde{g}$ (see point 1 above), I think the computation contains some mistakes. Most importantly, in equation (B.2), $\gamma$ should not be seen as the abstract segment $[0,1]$ but rather should be a path $\gamma_z : [0,1] \to U_q$ connecting $z$ with the boundary of $U_q$. In particular, this path depends on $z$ and the integration along it cannot be commuted with $\partial_z$ as done in (B.3). Since, this is also part of the main result of the paper yielding the unified gauged action, this should be derived very carefuly.

3) It seems to me that the boundary condition $y^{-1} \partial_z y |_q^{\mathfrak{h}} = 0$ imposed at a double pole $q$ in (4.33) allows to eliminate part of the degrees of freedom contained in the field $\Delta_q$. More precisely, the possibility to have $y^{-1} \partial_z y |_q$ take arbitrary values in $\mathfrak{f}$ (the complement of $\mathfrak{h}$ in $\mathfrak{g}$) allows to perform a gauge transformation that brings $g_q \Delta_q g_q^{-1}$ in the subalgebra $\mathfrak{h}$. This is consistent with the fact that the $\mathfrak{f}$-component of $g_q \Delta_q g^{-1}_q$ would eventually decouple from the gauged unified action (5.23), since $A_\pm|_q=B_\pm|_q\in\mathfrak{h}$. This point is less serious than the first 2 but in my opinion also impede an easy and complete understanding of the role of $\Delta_q$ and of the physical degrees of freedom of the resulting 2d theory.

---

## Round 5 · Author Response

Based on the comments provided by the referees I have made several changes to the paper. I've listed these in the box below.

---

## Round 5 · List of Changes

I have revised the following:
-Discussion of the regularisation of the action,
-Derivation of the valid gauge transformations and proof of their closure,
-Distinction between boundary conditions which are constraints, and those which are dynamical equations of motion,
-Relaxed the constraints imposed on the singular behaviour of the B-field. This allows for a larger class of field configurations and makes the role of the new scalars clear.

---

## Editorial Decision

in_refereeing